# RepLDM: Reprogramming Pretrained Latent Diffusion Models for High-Quality, High-Efficiency, High-Resolution Image Generation

**Boyuan Cao[1]**  **Jiaxin Ye[1]**  **Yujie Wei[1]**  **Hongming Shan[1]***

[1]Institute of Science and Technology for Brain-Inspired Intelligence &
MOE Key Laboratory of Computational Neuroscience and Brain-Inspired Intelligence &
MOE Frontiers Center for Brain Science, Fudan University
{caoby23, jxye22, yjwei22}@m.fudan.edu.cn, hmshan@fudan.edu.cn

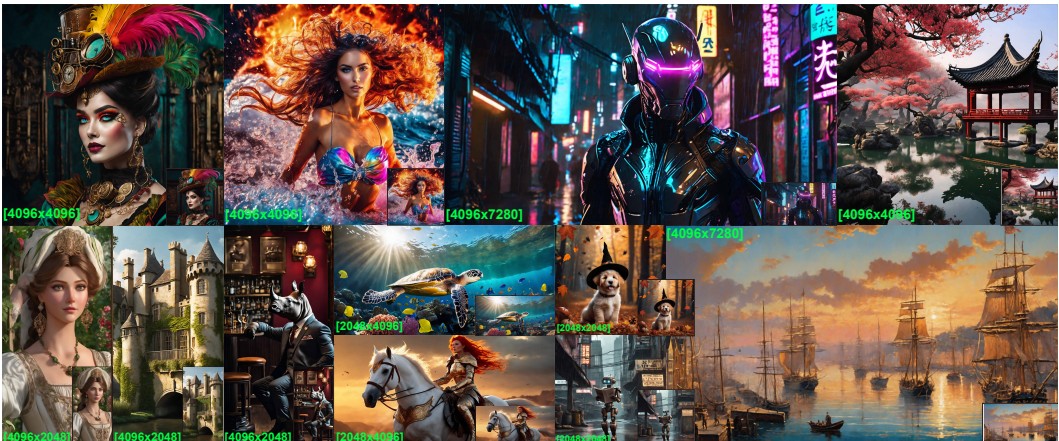

Figure 1: **High-resolution images generated by our RepLDM using a single consumer-grade 3090 GPU.** The corresponding thumbnails are generated by SDXL [36] at their training resolution.

## Abstract

While latent diffusion models (LDMs), such as Stable Diffusion, are designed for high-resolution (HR) image generation, they often struggle with significant structural distortions when generating images at resolutions higher than their training one. Instead of relying on extensive retraining, a more resource-efficient approach is to reprogram the pretrained model for HR image generation; however, existing methods often result in poor image quality and long inference time. We introduce RepLDM, a novel reprogramming framework for pretrained LDMs that enables *high-quality, high-efficiency, high-resolution* image generation; see Fig. 1. RepLDM consists of two stages: (**i**) an attention guidance stage, which generates a latent representation of a higher-quality training-resolution image using a novel training-free self-attention mechanism to enhance the structural consistency; and (**ii**) a progressive upsampling stage, which progressively performs upsampling in pixel space to mitigate the severe artifacts caused by latent space upsampling. The effective initialization from the first stage allows for denoising at higher resolutions with significantly fewer steps, improving the efficiency. Extensive experimental results demonstrate that RepLDM significantly outperforms state-of-the-art methods in both quality and efficiency for HR image generation, underscoring its advantages for real-world applications. Codes: https://github.com/kmittle/RepLDM.

---

*Corresponding author.

39th Conference on Neural Information Processing Systems (NeurIPS 2025).

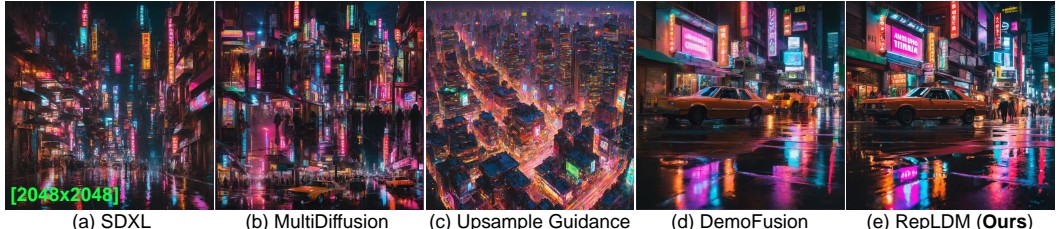

| (a) SDXL | (b) MultiDiffusion | (c) Upsample Guidance | (d) DemoFusion | (e) RepLDM (**Ours**) |

Figure 2: **Comparison of our RepLDM with prior work in generating 2048×2048 image**. The prompt is *Neon lights illuminate the bustling cityscape at night, casting colorful reflections on the wet streets*. Zoom-in for a better view.

## 1 Introduction

Diffusion models (DMs) have demonstrated impressive performance in visual generation tasks, particularly in text-to-image generation [7, 8, 16, 33, 34, 36, 46, 49–52, 55]. One notable variant of DMs is the latent diffusion model (LDM), which performs diffusion modeling in latent space to reduce training and inference costs, enabling HR generation up to $1024 \times 1024$. While it is plausible to modify the input size for higher-resolution generation, this often results in severe structural distortions, as illustrated in Fig. 2(a). Therefore, a recent research focus is on adapting trained LDMs for HR image generation without the need for additional training or fine-tuning (*i.e.* training-free manner), which can inherit the strong generation capacities of existing LDMs, especially open-sourced versions like Stable Diffusion.

Existing training-free approaches for HR image generation can be roughly categorized into three types: sliding window-based, parameter rectification-based, and progressive upsampling-based. Sliding window-based methods first divide the HR image into several overlapping patches and use sliding window strategies to perform denoising [1, 12, 25]. However, these methods could result in repeated structures and contents due to the lack of communication between windows; see Fig. 2(b). Parameter rectification-based methods attempt to correct models' parameters for better structural consistency through the entropy of attention maps, signal-to-noise ratio, and dilation rates of the convolution layers [14, 20–22, 56]. Though efficient, they often lead to the degradation of texture details; see Fig. 2(c). Unlike the two types mentioned above, progressive upscaling-based methods are to iteratively upsample the image resolution, which maintains better structural consistency and shows state-of-the-art (SOTA) performance [6, 27, 28, 37]. Unfortunately, these methods require fully repeating the denoising process multiple times, leading to an unaffordable computational burden; *e.g.*, AccDiffusion [28] takes 26 minutes to generate a $4096 \times 4096$ image. In addition, their upsampling operation in the latent space may introduce artifacts; see Fig. 2(d). To sum up, existing methods fail to ensure the fast, high-quality HR image generation.

In this paper, we propose RepLDM, a novel reprogramming framework for pretrained LDMs that is capable of generating high-quality, high-resolution images while keeping high-efficiency; see Fig. 2(e). Specifically, RepLDM decomposes the denoising process of LDMs into two stages: (**i**) an attention guidance stage, and (**ii**) a progressive upsampling stage. The first stage aims to generate a latent representation of a high-quality image at the training resolution through the proposed attention guidance, which is implemented via a novel training-free self-attention mechanism (TFSA) to improve structural consistency[2]. The second stage aims to progressively upsample the resolution in the pixel space rather than latent space, which can alleviate the severe artifacts caused by the latent space upsampling. By leveraging the effective initialization from the first stage, RepLDM can perform denoising in the second stage with significantly fewer steps, enhancing the overall efficiency with $5\times$ speedup. Extensive experimental results demonstrate the effectiveness and efficiency of RepLDM in generating HR images over the SOTA baselines.

The contributions of this work are summarized as follows. (**i**) We propose RepLDM, a novel framework for high-quality, high-efficiency, high-resolution image generation through reprogramming pretrained LDMs. (**ii**) We propose attention guidance, which can utilize a novel training-free self-

---

[2]In this paper, *structural consistency* refers to the plausibility of the overall scene layout and the realism of object structures within an image. Specifically, a reasonable layout should follow logical spatial relationships—for example, the sky should appear above the ground—while realistic object structures should conform to common sense, such as a cat having four legs rather than five.

attention to improve the structural consistency of the latent representation towards high-quality images at the training resolution. (**iii**) We propose progressively upsampling the resolution of latent representation in the pixel space, which can alleviate the artifacts caused by the latent space upsampling. (**iv**) Extensive experimental results demonstrate that the proposed RepLDM significantly outperforms the SOTA models in terms of image quality and inference time, emphasizing its great potential for real-world applications.

## 2   Related Work

**HR image generation with super-resolution.**   An intuitive approach to generating HR images is to first use a pre-trained LDM to generate training-resolution[3] (TR) images and then apply a super-resolution model to perform upsampling [26, 31, 47, 48, 54]. Although one can obtain structurally consistent HR images in this way, super-resolution models are primarily focused on enlarging the image, and shown to be unable to produce the details that users expect in HR images [6, 27, 28].

**HR image generation with additional training.**   Existing additional training methods either fine-tune existing LDMs with HR images [10, 19, 57] or train cascaded diffusion models to gradually synthesize higher-resolution images [17, 44]. Though effective, these methods require expensive training resources that are unaffordable for regular users.

**HR image generation in training-free manner.**   Current training-free methods can be roughly classified into three categories: sliding window-based, parameter rectification-based, and progressive upsampling-based methods. Sliding window-based methods consider spatially splitting HR image generation [1, 12, 25]. Specifically, they partition an HR image into several patches with overlap, and then denoise each patch. However, due to the lack of communication between windows, these methods result in structural disarray and content duplication. While enlarging the overlaps of the windows mitigates this issue, it can result in unbearable computational costs. For the parameter rectification-based methods, some researchers discovered that the collapse of HR image generation is due to the mismatches between higher resolutions and the model's parameters [14, 20–22, 56]. These methods attempt to eliminate the mismatches by rectifying the parameters such as the dilation rates of some convolutional layers. While mitigating the structural inconsistency, they often lead to the degradation of image details. Different from the aforementioned two types, the progressive upsampling-based methods show SOTA performance in some recent studies [6, 27, 28, 37]. Though promising, they require fully repeating the denoising process multiple times, which incurs unbearable computational overhead. Additionally, these methods perform upsampling in the latent space, which may introduce artifacts.

Although their remarkable results, these methods fail to improve the quality of HR images and computational efficiency at the same time. In contrast, RepLDM aims to generate HR images with high quality and high efficiency, towards practical applications.

## 3   Method

### 3.1   Overview of RepLDM

Fig. 3 presents the overview of RepLDM, which reprograms a pre-trained LDM to generate HR images without further training. Formally, a pre-trained LDM utilizes a denoising U-Net model $\mathcal{F}$ to iteratively denoise the latent representation of size $h \times w \times c$, which is then converted back to the pixel space for final image generation through the decoder $\mathcal{D}$ of a variational autoencoder (VAE). We note that the initial latent representation is sampled from a Gaussian distribution $\epsilon \sim \mathcal{N}(0, \boldsymbol{I})$, and for inference the encoder $\mathcal{E}$ of VAE is not involved.

Our RepLDM extends pre-trained LDMs for higher-resolution image generation in a training-free manner; *i.e.*, $\mathcal{E}$, $\mathcal{D}$ and $\mathcal{F}$ are fixed. RepLDM achieves this by decomposing the standard denoising process in the latent space into two stages: (**i**) attention guidance stage, and (**ii**) progressive upsampling stage. In the first stage, RepLDM aims to generate a latent representation of a higher-quality TR

---

[3]In this paper, *training resolution* refers to the resolution used during model training, while *high resolution* denotes a resolution that substantially exceeds the training resolution—beyond the level at which the model can directly produce satisfactory results.

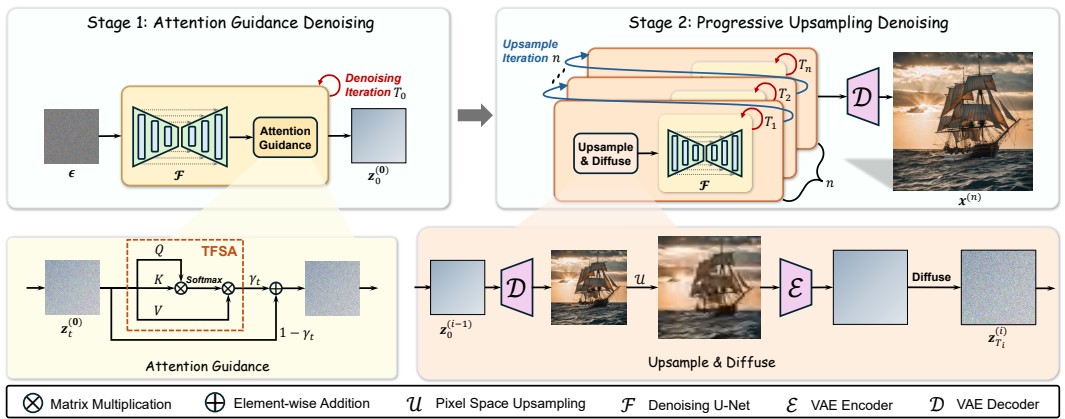

Figure 3: **Overview of RepLDM**. RepLDM divides the denoising process of a pre-trained LDM into two stages. The first stage leverages the introduced attention guidance to enhance the structural consistency by utilizing a novel training-free self-attention mechanism (TFSA). The second stage iteratively upsamples the latent representation in pixel space to eliminate artifacts.

image through the proposed attention guidance. The attention guidance is implemented as linearly combining the novel training-free self-attention mechanism (TFSA) and original latent representation to improve the structural consistency. In the second stage, RepLDM uses the latent representation provided by the first stage as a better initialization, and iteratively obtains higher-resolution images via the pixel space upsampling and diffusion-denoising refinement.

We detail the attention guidance stage in §3.2, followed by the progressive upsampling stage in §3.3.

## 3.2 Attention Guidance Stage

**Motivation.** Enhancing the structural consistency helps improve image quality [43]. However, it is challenging to do this in a training-free manner. We observe that the self-attention mechanism presents powerful global spatial modeling capability [5, 13, 29, 45], and this capability is parameter-agnostic. It is determined by the paradigm of global similarity calculation inherent to the self-attention mechanism [45, 58]. These insights motivate us to consider designing a novel training-free self-attention mechanism to elegantly enhance the global structural consistency of the latent representation.

**Denoising with attention guidance.** To improve the structural consistency of the latent representation at the training resolution $z \in \mathbb{R}^{h \times w \times c}$, we propose a simple yet effective training-free self-attention mechanism for attention guidance, termed TFSA, formulated as:

$$\text{TFSA}(z) = f^{-1} \left( \text{Softmax} \left( \frac{f(z)f(z)^{\text{T}}}{\lambda} \right) f(z) \right), \tag{1}$$

where the operation $f$ reshapes the latent representation into shape $(hw) \times c$ and $f^{-1}$ reshapes it back; $\lambda$ is the scaling factor, with a default value of $\lambda = \sqrt{c}$.

However, we empirically observe that directly using the TFSA in Eq. (1) to improve the structural consistency of the latent representation could lead to unstable denoising. Therefore, we propose linearly combining the outputs of TFSA and the original latent representation as attention guidance, which is formulated as:

$$\tilde{z} = \gamma \text{TFSA}(z) + (1 - \gamma) z, \tag{2}$$

where $\tilde{z}$ is the structurally enhanced latent representation and $\gamma$ is the guidance scale. In Appendix A, we demonstrate that TFSA functions by modulating the distribution of latent representations. The term $(1-\gamma)z$ in Eq. (2) serves as a statistical anchor, helping to keep the guided latent representations on the data manifold and ensuring smooth transitions in their distribution.

As shown in Fig. 3, we append the attention guidance in Eq. (2) to denoising U-Net model $\mathcal{F}$ and repeat the denoising process for a total of $T_0$ times for the first stage. We note that the denoising process starts from step $T_0$ to 1, and the final output of the first stage is denoted as $z_0^{(0)}$.

**Adaptive guidance scale.** Considering that the latent representation is mostly non-semantic noise in the first few steps of denoising, we delay $k$ steps in introducing attention guidance. Moreover,

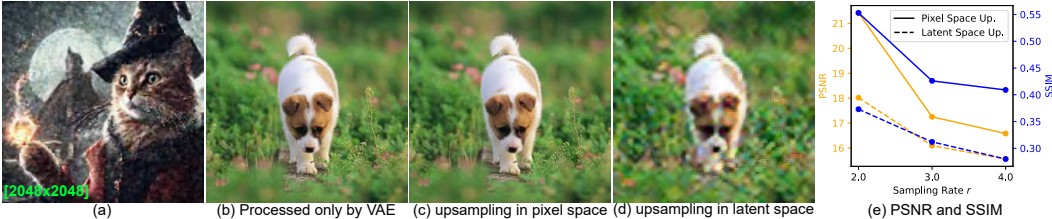

| (a) | (b) Processed only by VAE | (c) upsampling in pixel space | (d) upsampling in latent space | (e) PSNR and SSIM |

Figure 4: **Comparison between upsampling in pixel space and latent space.** (a) RepLDM with latent space upsampling leads to severe artifacts. (b)-(e): Qualitative and quantitative comparisons of different upsampling methods.

during the denoising process, the image structure is generated first, followed by local details [32, 44, 53]. Therefore, we primarily employ attention guidance in the early to mid-steps of denoising to focus on enhancing the structural consistency of the latent representation. Specifically, we introduce the adaptive guidance scale $\gamma_t$ by applying a decay to a given guidance scale $\gamma$, formulated as:

$$\gamma_t = \begin{cases} \gamma\left[\frac{1}{2}\left[\cos\left(\frac{T_0-k-t}{T_0-k}\pi\right)+1\right]\right]^\beta & \text{if } t \leq T_0 - k, \\ 0 & \text{otherwise,} \end{cases} \tag{3}$$

where $\beta$ is the decay factor. In practice, considering that $k$ depends on $T_0$ for different resolutions, we use a delay rate $\eta_1 = \frac{k}{T_0}$ to control the number of steps for delaying attention guidance.

### 3.3 Progressive Upsampling Stage

**Motivation.** Fig. 2(a) shows that pre-trained LDMs still retain some ability to generate high-frequency information when directly used to synthesize HR images, although they exhibit structural disarray. Therefore, intuitively, we can utilize the latent representation produced by the first stage as a structural initialization, and generate the HR images through the "upsample-diffuse-denoise" iteration in the latent space. However, this pipeline leads to severe artifacts, as shown in Fig. 4(a). We speculate that this is due to *the upsampling of latent representations in the latent space*.

**Pilot study.** To examine this hypothesis, we conduct the following experiments. Specifically, we randomly select 10k images from ImageNet [4] to create an image set $\mathcal{P}$. For each image $\boldsymbol{x} \in \mathcal{P}$, we perform the following operations to obtain three additional image sets: (i) $\hat{\boldsymbol{x}} = \mathcal{D} \circ \mathcal{E}(\boldsymbol{x})$, which use VAE to obtain the reconstructed image set $\mathcal{P}_{\text{ref}}$; (ii) $\hat{\boldsymbol{x}} = \text{up} \circ \mathcal{D} \circ \mathcal{E} \circ \text{down}(\boldsymbol{x})$, which performs upsampling in pixel space to obtain the image set $\mathcal{P}_{\text{pix}}$; and (iii) $\hat{\boldsymbol{x}} = \mathcal{D} \circ \text{up} \circ \mathcal{E} \circ \text{down}(\boldsymbol{x})$, which performs upsampling in latent space to obtain the image set $\mathcal{P}_{\text{lat}}$. Both upsampling up and downsampling down are performed using bicubic interpolation. Fig. 4(e) reports the quantitative results, where $r$ represents the upsampling or downsampling rate. We calculate the PSNR and SSIM for pixel space upsampling set $\mathcal{P}_{\text{pix}}$ and latent space upsampling set $\mathcal{P}_{\text{lat}}$ with respective to the reference set $\mathcal{P}_{\text{ref}}$. It can be clearly observed that the latent space upsampling leads to a significant performance decline compared to pixel space upsampling. Fig. 4(b-d) shows upsampling in the pixel space produces images close to the reference while upsampling in latent space leads to severe artifacts and detail loss.

**Progressive denoising with pixel space upsampling.** Based on the above conclusion, we propose performing upsampling in the pixel space rather than latent space and utilize diffusion and denoising to refine the upsampled higher-resolution image. Specifically, the second stage consists of $n$ sub-stages to progressively upsample the training resolution to target resolution, each corresponding to one upsampling operation. For $i$-th sub-stage, $i = 1, \dots, n$, we prepend an upsample and diffuse operation before the denoising process, which can be defined as:

$$\begin{aligned} \hat{\boldsymbol{z}}_0^{(i-1)} &= \mathcal{E} \circ \mathcal{U} \circ \mathcal{D}(\boldsymbol{z}_0^{(i-1)}), \\ \boldsymbol{z}_{T_i}^{(i)} &= \sqrt{\bar{\alpha}_{T_i}}\hat{\boldsymbol{z}}_0^{(i-1)} + \sqrt{1-\bar{\alpha}_{T_i}}\boldsymbol{\epsilon}, \end{aligned} \tag{4}$$

where $\mathcal{U}$ represents upsampling operation, $\bar{\alpha}_{T_i}$ is the noise schedule hyper-parameter of the $T_i$-th diffusion time step, and $\boldsymbol{z}_0^{(i-1)}$ is the output of the $(i-1)$-th sub-stage; we use $\boldsymbol{z}_0^{(0)}$ to denote the output from the first stage. Then, $\mathcal{F}$ is used to iteratively denoise $\boldsymbol{z}_{T_i}^{(i)}$ from time step $T_i$ to obtain $\boldsymbol{z}_0^{(i)}$.

After completing all sub-stages, we obtain $z_0^{(n)}$, which is then decoded to produce the final output $\boldsymbol{x}^{(n)} = \mathcal{D}(\boldsymbol{z}_0^{(n)})$.

We empirically found that generating higher-resolution images requires more sub-stages. Additionally, when refining images using diffusion and denoising, higher resolutions demand larger time steps [44]. In practice, for flexibility, RepLDM allows users to customize the number of sub-stages $n$, and the diffusion time steps $T_i$ for each sub-stage by a pre-specified variable-length progressive scheduler $\eta_2 = \left[\frac{T_1}{T_0}, \frac{T_2}{T_0}, \ldots, \frac{T_n}{T_0}\right]$. The elements of $\eta_2$ represent the denoising steps of each sub-stage, normalized by $T_0$.

# 4 Experiments

## 4.1 Implementation Details

**Experimental settings.** We use SDXL [36] as the pre-trained LDM and conduct inference using a single NVIDIA 4090 GPU. To ensure consistency when testing inference speed, we use a single 3090 GPU, aligning with other methods. We randomly sample 33k images from the segment anything model (SAM) [24] dataset as the benchmark. Following the released code from DemoFusion [6], we use the EulerDiscreteScheduler [23] setting $T_0 = 50$ and the classifier-free guidance [18] scale to 7.5. Pixel space upsampling is performed using bicubic interpolation, and the decay factor $\beta$ is fixed at 3.

**Evaluation metrics.** The widely recognized metrics Frechet Inception distance (FID) [15], Inception score (IS) [40], and contrastive language-image pre-training (CLIP) score [38] are used to evaluate model performance. Additionally, since calculating FID and IS requires resizing images to $299 \times 299$, which may not be suitable for evaluating HR images, we follow the experimental settings of [6, 28] to perform ten $1024 \times 1024$ window crops on each image to calculate $FID_c$ and $IS_c$. Since FID is known to be sensitive to small implementation details [35], we employ a widely recognized implementation from a publicly available repository [42].

## 4.2 Quantitative Results

We compare RepLDM with the following models: (1) SDXL [36]; (2) MultiDiffusion [1]; (3) ScaleCrafter [14]; (4) DemoFusion [6]; (5) Upsample Guidance (UG) [21]; (6) AccDiffusion [28]; and (7) HiDiffusion [56]. For fair comparisons, we disabled the FreeU trick [43] in all experiments.

Table 1: **Quantitative comparison results**. The best results are marked in **bold**, and the second best results are marked by underline.

| Method | 2048 × 2048 | | | | | 2048 × 4096 | | | | | 4096 × 2048 | | | | | 4096 × 4096 | | | | |
|---|---|---|---|---|---|---|---|---|---|---|---|---|---|---|---|---|---|---|---|---|
| | FID | IS | $FID_c$ | $IS_c$ | CLIP | FID | IS | $FID_c$ | $IS_c$ | CLIP | FID | IS | $FID_c$ | $IS_c$ | CLIP | FID | IS | $FID_c$ | $IS_c$ | CLIP |
| SDXL [36] | 99.9 | 14.2 | 80.0 | 16.9 | 25.0 | 149.9 | 9.5 | 106.3 | 12.0 | 24.4 | 173.1 | 9.1 | 108.5 | 11.5 | 23.9 | 191.4 | 8.3 | 114.1 | 12.4 | 22.9 |
| MultiDiff. [1] | 98.8 | 14.5 | 67.9 | 17.1 | 24.6 | 125.8 | 9.6 | 71.9 | 15.7 | 24.6 | 149.0 | 9.0 | 70.5 | 14.4 | 24.4 | 168.4 | 6.5 | 76.6 | 14.4 | 23.1 |
| ScaleCrafter [14] | 98.2 | 14.2 | 89.7 | 13.3 | 25.4 | 161.9 | 10.0 | 154.3 | 7.5 | 23.3 | 175.1 | 9.7 | 167.3 | 8.0 | 21.6 | 164.5 | 9.4 | 170.1 | 7.3 | 22.3 |
| UG [21] | 82.2 | 17.6 | 65.8 | 14.6 | **25.5** | 155.7 | 8.2 | 165.0 | 6.6 | 21.7 | 185.3 | 6.8 | 175.7 | 6.2 | 20.5 | 187.3 | 7.0 | 197.6 | 6.3 | 21.8 |
| HiDiff. [56] | 81.0 | 16.8 | 64.1 | 14.2 | 24.9 | 120.7 | 12.2 | 93.0 | 13.6 | 24.2 | 128.4 | 12.8 | 98.3 | 11.3 | 23.1 | 144.1 | 12.5 | 147.0 | 7.4 | 21.2 |
| DemoFusion [6] | 72.3 | **21.6** | 53.5 | **19.1** | 25.2 | 96.3 | 17.7 | 62.3 | 15.0 | **25.0** | 99.6 | 16.4 | 61.9 | 14.7 | 24.4 | 101.4 | 20.7 | 63.5 | 13.5 | **24.7** |
| AccDiff. [28] | 71.6 | 21.0 | 52.7 | 17.0 | 25.1 | **95.5** | 16.4 | 62.9 | 11.1 | 24.5 | 102.2 | 15.2 | 65.4 | 11.5 | 24.2 | 103.2 | 20.1 | 65.9 | 13.3 | 24.6 |
| RepLDM | **66.0** | 21.0 | **47.4** | 17.5 | 25.1 | 89.0 | 20.3 | 56.0 | 19.0 | 25.0 | 93.2 | 19.5 | 56.9 | 16.5 | 24.9 | 90.6 | 21.1 | 59.0 | 14.8 | 24.6 |

We report the performance of all methods on four different resolutions (Height $\times$ Width): $4096 \times 4096$, $4096 \times 2048$, $2048 \times 4096$, and $2048 \times 2048$. Considering that the generation time for HR images far exceeds that for low-resolution images, we used 2k prompts at the resolution of $2048 \times 2048$, and 1k prompts for resolutions greater $2048 \times 2048$. For all resolutions, we set $\gamma = 0.004$, $\beta = 3$ and $\eta_1 = 0.06$ for RepLDM. Given that the $4096 \times 4096$ resolution is significantly larger than other resolutions, we set $\eta_2 = [0.1, 0.2]$ (*i.e.*, $T_0 = 50$, $T_1 = 5$, $T_2 = 10$) for $4096 \times 4096$, and $\eta_2 = [0.2]$ (*i.e.*, $T_0 = 50$ and $T_1 = 10$) for other resolutions. When generating images with an aspect ratio of $r'$, we reshape the initially sampled Gaussian noise $\epsilon$ in the first stage to match $r'$. This process keeps the number of tokens in $\epsilon$ unchanged, preventing drastic fluctuations in the entropy of the attention maps in the transformer [22] leading to higher-quality images.

Table 1 manifests that RepLDM significantly outperforms previous SOTA models, AccDiffusion and DemoFusion. This indicates that RepLDM generates images with higher quality. For more comprehensive analyses, we repeat the experiments of Table 1 with different random seeds to perform error analyses and conduct a further comparison of the models on the LAION-5B benchmark [41]; see Appendix G.

Table 2 indicates that RepLDM demonstrates remarkable advantage in inference speed compared to the SOTA models. On a single 3090 GPU, RepLDM requires only about one-fifth of the inference time needed by SOTA models such as DemoFusion and AccDiffusion.

Table 2: **Model inference time**. The best results are marked in **bold**. Unit of Time: minute.

| Resolutions | SDXL [36] | MultiDiff. [1] | ScaleCrafter [14] | UG [21] | DemoFusion [6] | AccDiff. [28] | HiDiff. [56] | RepLDM |
|---|---|---|---|---|---|---|---|---|
| 2048 × 2048 | 1.0 | 3.0 | 1.0 | 1.8 | 3.0 | 3.0 | 0.8 | **0.6** |
| 2048 × 4096 | 3.0 | 6.0 | 6.0 | 4.0 | 11.0 | 12.7 | **1.9** | 2.0 |
| 4096 × 4096 | 8.0 | 15.0 | 19.0 | 11.1 | 25.0 | 26.0 | **3.4** | 5.7 |

## 4.3 Qualitative Results

In Fig. 5, RepLDM is qualitatively compared with AccDiffusion, DemoFusion, and MultiDiffusion. MultiDiffusion fails to maintain global semantic consistency. As indicated by the red boxes, Demo-Fusion and AccDiffusion tend to result in chaotic content repetition and severe artifacts, which we speculate are caused by upsampling in the latent space (as analyzed in §3.3). In contrast, RepLDM not only preserves excellent global structural consistency but also synthesizes images with more details. More qualitative comparison results can be found in Appendix B.

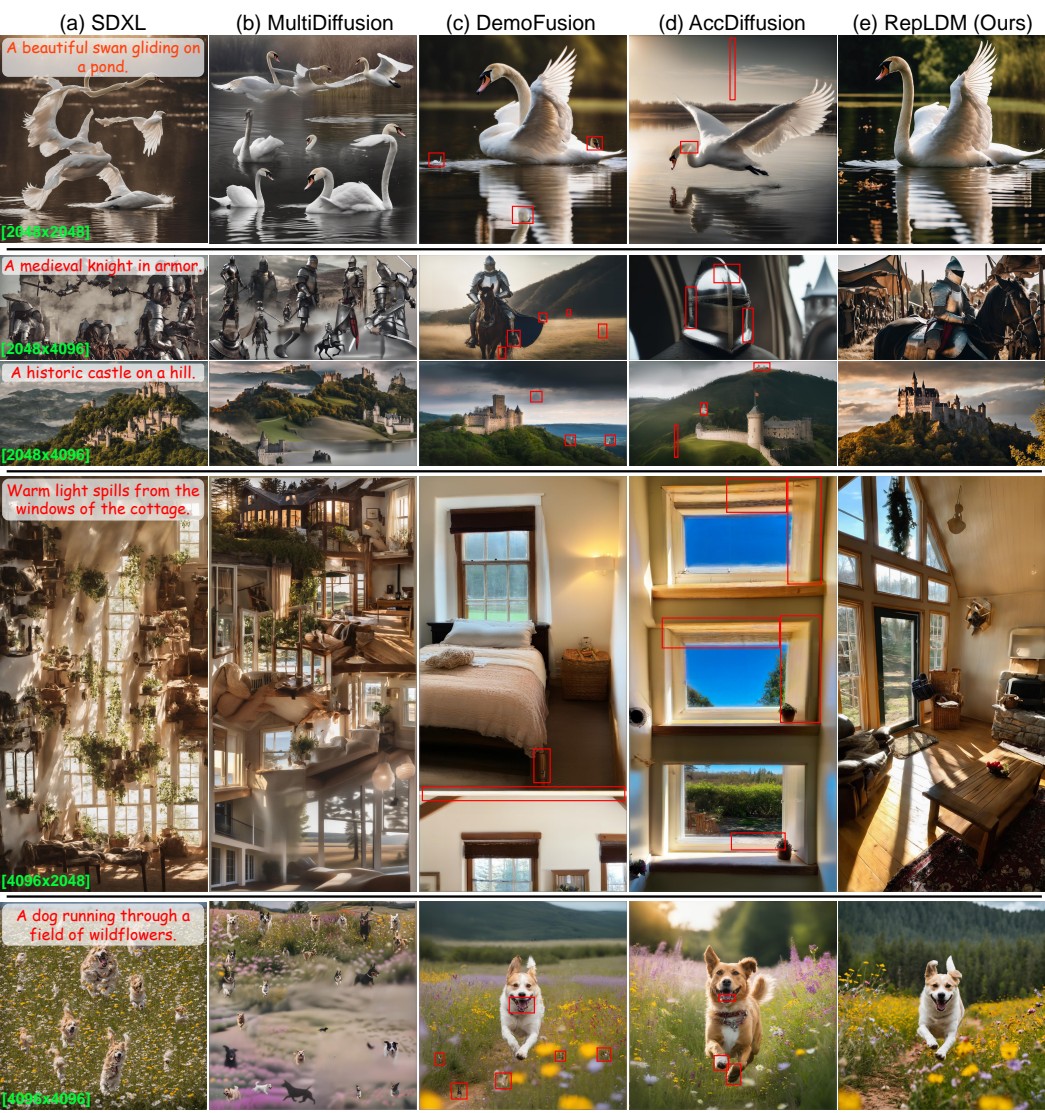

Figure 5: **Qualitative comparison with other baselines.** The prompts used to generate the images are presented in the white boxes. MultiDiffusion fails to maintain global semantic consistency. DemoFusion and AccDiffusion exhibit severe artifacts and content repetition. The red boxes indicate some synthesis errors. Zoom-in for a better view.

## 4.4 User Study

We invite 16 volunteers to participate in a double-blind experiment to further evaluate the performance of the models. Each volunteer is required to answer 35 questions. In each question, three images gener-

Table 3: **Results of the user study.**

| Method | Structural Consistency | | Color Abundance | | Detail Richness | |
|---|---|---|---|---|---|---|
| | score ↑ | score* ↑ | score ↑ | score* ↑ | score ↑ | score* ↑ |
| AccDiff. [28] | 6.28 | 0.88 | 6.78 | 0.60 | 6.18 | 0.53 |
| DemoFusion [6] | 5.99 | 0.59 | 6.69 | 0.51 | 6.18 | 0.53 |
| RepLDM | **7.42** | **2.02** | **7.64** | **1.45** | **7.41** | **1.76** |

ated by AccDiffusion, DemoFusion, and RepLDM are presented. The volunteer needs to rate each image from 1 to 10 in terms of structural consistency, color abundance, and detail richness. We calculate the average of their scores. Moreover, to eliminate bias in each volunteer's ratings for each metric in each question, we subtract the minimum value among the three scores given by each volunteer for each metric in each question. The rectified score is denoted as score*. Table 3 shows that RepLDM surpasses previous SOTA models across all metrics.

## 5 Ablation Study

### 5.1 Attention Guidance

In this section, we first conduct ablation experiments on attention guidance, followed by ablation experiments on the hyper-parameters of attention guidance. In Appendix A, we provide a detailed analysis of how attention guidance improves latent structural consistency and image quality.

**Ablation on attention guidance.** We keep $\eta_2$ unchanged and analyze the effect of attention guidance through qualitative and quantitative experiments. Table 4 shows that attention guidance leads to improvements across various metrics, indicating that using attention guidance to enhance the consistency of latent encoding results in higher-quality images. The qualitative experiments in Fig. 6 demonstrate that using attention guidance eliminates image blurriness and enriches the image details. Note that FID and IS quantify the statistical differences between two distributions [2, 15, 40]. Since attention guidance mainly enhances visual quality by modifying the mid- and high-frequency components while preserving the low-frequency structure of the image, it has limited impact on the overall distributional statistics. Although attention guidance may not yield significant improvements in quantitative metrics, it provides a noticeable enhancement in human visual perception; see Table 3. Please refer to Appendix C.2 for additional qualitative ablation results.

Table 4: **Ablation on attention guidance (AG).** The best results are marked in **bold**.

| Method | 2048 × 2048 | | | | 2048 × 4096 | | | | 4096 × 2048 | | | | 4096 × 4096 | | | |
|---|---|---|---|---|---|---|---|---|---|---|---|---|---|---|---|---|
| | FID | IS | FID$_c$ | IS$_c$ | CLIP | FID | IS | FID$_c$ | IS$_c$ | CLIP | FID | IS | FID$_c$ | IS$_c$ | CLIP | FID | IS | FID$_c$ | IS$_c$ | CLIP |
| w/o AG | 66.8 | **21.6** | 47.5 | 17.4 | **25.3** | 91.6 | **20.3** | 58.0 | 14.5 | **25.0** | 95.3 | **19.9** | 58.4 | 14.5 | **24.9** | 92.0 | **21.6** | 59.8 | 13.6 | 24.5 |
| w/ AG | **66.0** | 21.0 | **47.4** | **17.5** | 25.1 | **89.0** | 20.3 | **56.0** | **19.0** | **25.0** | **93.2** | 19.5 | **56.9** | **16.5** | **24.9** | **90.6** | 21.1 | **59.0** | **14.8** | 24.6 |

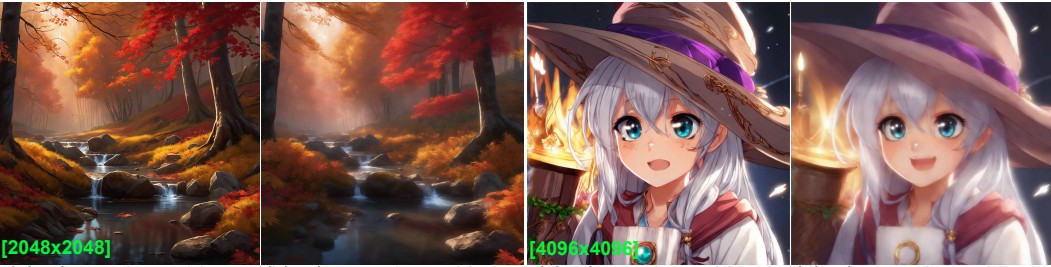

(a) w/ attention guidance  (b) w/o attention guidance  (c) w/ attention guidance  (d) w/o attention guidance

Figure 6: **Ablation on attention guidance**. Zoom-in for a better view.

**Ablation on attention guidance with ControlNet.** To further demonstrate the generalization ability of attention guidance, in this section, we perform an qualitative ablation study of attention guidance with ControlNet [55]. Specifically, we conducted comparative experiments using two types of conditional guidance (canny and depth) across two resolution scales: 4096 × 4096 and 2048 × 2048. As shown in Fig. 7, the integration of attention guidance with ControlNet substantially enhances chromatic fidelity and structural granularity in synthesized images.

**Ablation on guidance scale $\gamma$.** We fix $\eta_1 = 0.06, \eta_2 = [0.2]$ and then explore the effect of the guidance scale $\gamma$ through both quantitative and qualitative experiments. For the quantitative

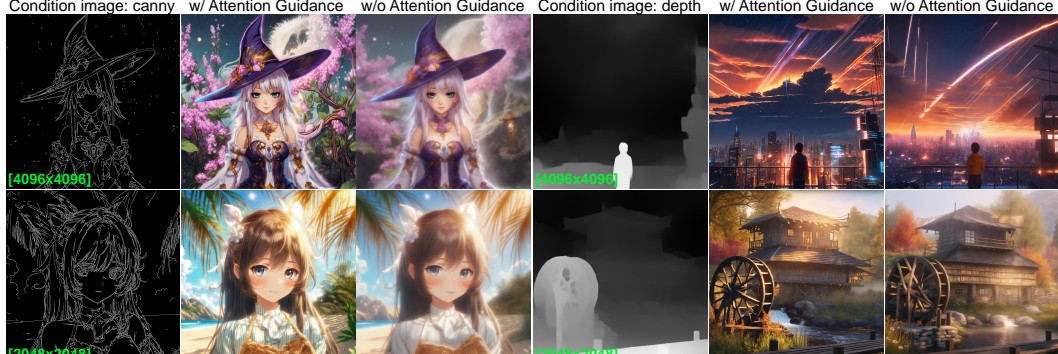

Figure 7: **Ablation on attention guidance with ControlNet.**

experiments, we find that $\gamma = 0.004$ performs better. Interestingly, when a larger $\gamma$ is used, the visual quality of the images can be further enhanced. As shown in Fig. 8, using a larger guidance scale results in richer image details. This allows users to generate images according to their preferences for detail richness and color contrast by adjusting the guidance scale. The setup and results of the quantitative experiments are detailed in Appendix C.2.

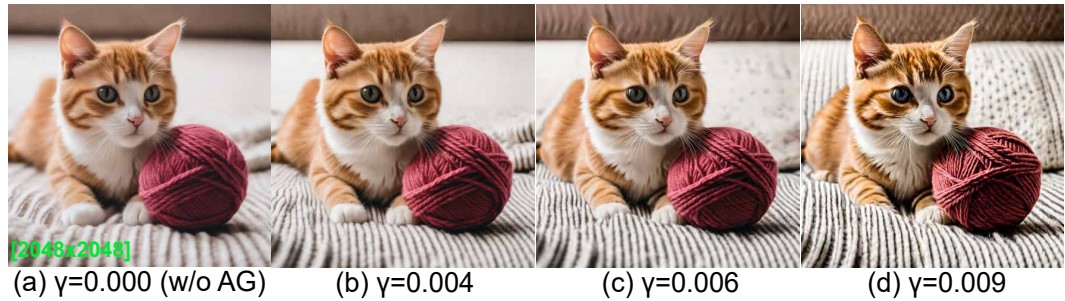

(a) γ=0.000 (w/o AG)     (b) γ=0.004     (c) γ=0.006     (d) γ=0.009

Figure 8: **Ablation on guidance scale**. Zoom-in for a better view.

**Ablation on delay rate $\eta_1$.** We fix $\gamma = 0.004, \eta_2 = [0.2]$ and then investigate the impact of the delay rate $\eta_1$ through both quantitative and qualitative experiments. The quantitative analysis results indicate that better generation results can be achieved when $\eta_1 = 0.06$, indicating that appropriately delaying the effect of attention guidance contributes to further improving the quality of the images.

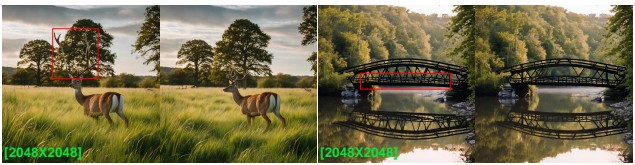

(a) $\eta_1 = 0.00$   (b) $\eta_1 = 0.06$   (c) $\eta_1 = 0.00$   (d) $\eta_1 = 0.06$

Figure 9: **Ablation on delay rate.** Errors indicated by red boxes can be eliminated by delaying attention guidance. Zoom-in for a better view.

We conjecture that this is because, at the very beginning of the denoising process, the structural information in the latent encoding has not yet emerged, and thus attention guidance cannot effectively enhance structural consistency. As shown in Fig. 9, delaying the effect of attention guidance eliminates some generation errors, further improving image quality. The setup and results of the quantitative experiments are detailed in Appendix C.2.

**Ablation on the time steps of attention guidance.** To explain why attention guidance needs to be applied during the early to middle steps of denoising, we apply attention guidance during different denoising steps of the first stage: (a) 47 to 33, (b) 32 to 17, and (c) 16 to 1. Fig. 10 shows that when attention guidance is applied

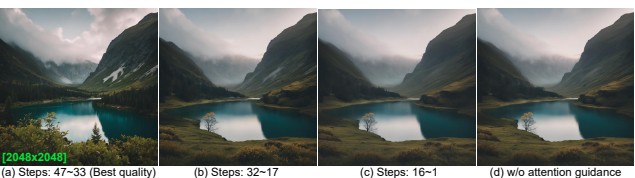

(a) Steps: 47~33 (Best quality)   (b) Steps: 32~17   (c) Steps: 16~1   (d) w/o attention guidance

Figure 10: **Applying attention guidance at different denoising steps.** Zoom-in for a better view.

during the early to middle steps of denoising, the image becomes clearer and more detailed; however, when attention guidance is applied during the later steps of denoising, it has negligible effect on the generated image. We speculate that this is because diffusion models tend to synthesize

structural information first [32, 44, 53], and once the structural information is generated, attention guidance may have a limited impact on structural consistency.

## 5.2 Progressive High-Resolution Denoising

In this section, we conduct ablation experiments on the progressive scheduler $\eta_2$ in the second stage of RepLDM. Specifically, we fixed $\gamma = 0, \eta_1 = 0$ and then explore the effect of the progressive scheduler $\eta_2$ through both quantitative and qualitative experiments. Quantitative experimental results indicate that an excessively large progressive scheduler value may result in a decline in image quality. This can also be observed in

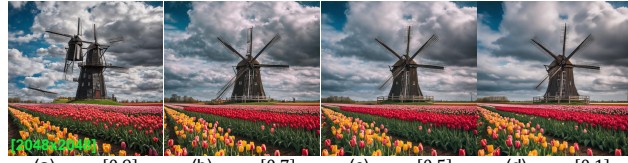

(a) $\eta_2 = [0.9]$    (b) $\eta_2 = [0.7]$    (c) $\eta_2 = [0.5]$    (d) $\eta_2 = [0.1]$

Figure 11: **Generated images using different** $\eta_2$**.** (a): When the value of progressive scheduler is too large, the structural repetition issue may reappear. (b) to (d): The visual effects are similar. Therefore, we can use a smaller progressive scheduler value to accelerate inference.

Fig. 11. It is evident that a too large progressive scheduler value may lead to structural misalignment and repetition issues observed in pre-trained SDXL. When the progressive scheduler value is sufficiently small, changing it yields similar visual effects. Therefore, we can choose a smaller progressive scheduler value (*e.g.*, 0.2) to accelerate inference. The setup and quantitative results are detailed in Appendix C.2.

## 6 Limitations And Future Work

RepLDM exhibits limitations in the following aspects: (**i**) Effectively controlling text in images is challenging, as demonstrated by examples in Fig. 12. This may be due to the inherent limitations of SDXL in generating textual symbols. Text, due to its more regular structure compared to other image content, is difficult to restore by directly enhancing the structural consistency of the latent representation. We speculate that the most

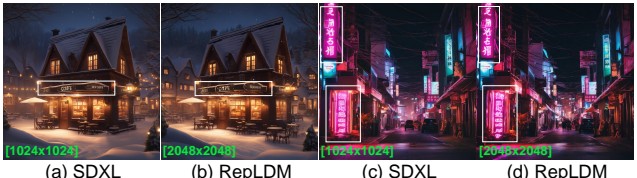

(a) SDXL    (b) RepLDM    (c) SDXL    (d) RepLDM

Figure 12: **Limitations of RepLDM.** The generation results of SDXL at its training resolution and those of RepLDM at higher resolutions are provided. As indicated by the white boxes, RepLDM fails to address the text structure errors inherited from SDXL.

reliable approach would be to fine-tune the model specifically on images containing text. (**ii**) When generating ultra-high resolution images, such as $12800 \times 12800$, the second stage of RepLDM inevitably needs to be decomposed into more sub-stages, which increases the model's inference time.

Developing a low-cost and effective fine-tuning method to correct text generation errors may be a promising topic. Moreover, adapting attention guidance to other tasks, such as video generation can be an interesting issue.

## 7 Conclusion

In this paper, we reprogram pretrained LDMs, unlock their potentials, and propose RepLDM for high-quality, high-efficiency, high-resolution image generation. RepLDM divides the denoising process of an LDM into two stages: (**i**) attention guidance stage, and (**ii**) progressive upsampling stage. The first stage generates structurally enhanced latent representations through the proposed attention guidance, employing a novel parameter-free self-attention mechanism. The second stage iteratively performs upsampling in the pixel stage, thus eliminating the artifacts caused by latent space upsampling. Extensive experiments show that our proposed RepLDM significantly outperforms SOTA models while achieving $5\times$ speedup in HR image generation.

## Acknowledgement

This work was supported by the National Natural Science Foundation of China (No. 62471148) and the computations in this research were supported by the CFFF platform of Fudan University.

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

# Appendix

## A    How Does TFSA/Attention Guidance Work?

In this section, we further elaborate on the working mechanism of attention guidance. Our attention guidance enhances the structural consistency of the latent representation by integrating the output of TFSA. Therefore, we conduct a detailed analysis of TFSA. Specifically, the functionality of TFSA can be described in two aspects: (**i**) *clustering the related tokens* in the latent representations; (**ii**) *adjusting the amplitude of the high-frequency and low-frequency components* in the latent representations.

### A.1    TFSA Clusters Semantically Related Tokens

**Visualization of the clustering effect of TFSA.**    TFSA reorganizes tokens based on their similarities. Intuitively, this enables TFSA to perform token clustering, which enhances the structural

consistency of latent representations. To demonstrate the clustering effect of TFSA, we calculated the deviation of the tokens' mean (DTM) of the latent representations $\tilde{z}_t$ and $z_t$. Concretely, assuming $z_t \in \mathbb{R}^{h \times w \times c}$, and $Z_t = \text{Flatten}(z_t) = [y_{t1}, \ldots, y_{tN}] \in \mathbb{R}^{N \times c}$, where $N = h \times w$, we calculate DTM as:

$$\text{DTM} = [\text{mean}(y_{ti}) - \text{mean}(Z_t) \text{ for } i = 1, \ldots, N] \tag{5}$$

To provide an intuitive illustration of the clustering effect of TFSA, we visualize the DTM based on token indices (*i.e.*, $i = 1, \ldots, N$) when $t$ is relatively large. As shown in columns (A) and (B) of Fig. 13, compared to the DTM of $z_t$ (blue points), the DTM of $\tilde{z}_t$ (red points) becomes more dispersed and exhibits distinct stripe patterns, indicating that TFSA indeed clusters the tokens of the latent representations. This clustering effect can be more directly demonstrated when $t$ is smaller. As shown in the heatmaps in columns (C) and (D) of Fig. 13, it is evident that TFSA clusters semantically related tokens.

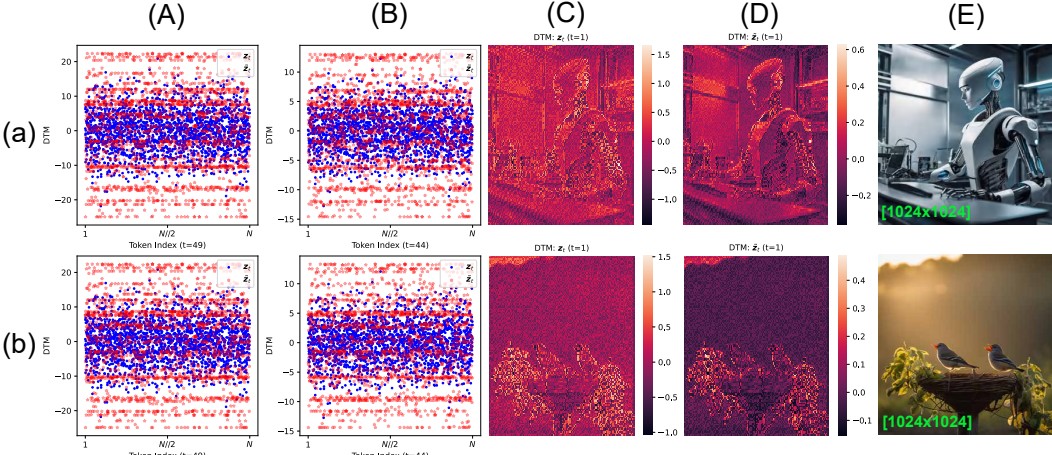

Figure 13: **The clustering effect of TFSA.** Columns (A), (B), (C), and (D) show the DTM of latent representations, while column (E) presents the corresponding generated RGB images.

**The clustering effect of TFSA leads to accelerated structural denoising.** Fig. 13 shows that the clustering effect of TFSA clarifies the semantic structures of objects, enabling the model to complete the denoising of low-frequency structures earlier. This early revelation of the overall image layout provides a stronger prior for subsequent fine-detail generation. To illustrate this, Fig. 14 presents the denoising process for the ablation of attention guidance. Note the regions highlighted by red boxes. With the incorporation of attention guidance, these areas exhibit clearer structures, which facilitates the generation of more affluent details and more vivid colors in subsequent steps.

To quantitatively demonstrate that TFSA accelerates structural emergence, we calculate the SSIM between $z_t$ and $z_0$, where $t \in 1, 2 \ldots, T-1$, and $T = 50$. As shown in Fig. 15, compared to the naive denoising process, attention guidance consistently drives the latent representations closer to their final states at each step, indicating the structural foreseeability of TFSA.

## A.2 TFSA Adjusts the Amplitude of High- and Low-frequency Components

The aim of this experiment is to explain: (**i**) why appropriately delaying attention guidance can resolve structural deformation issues (as shown

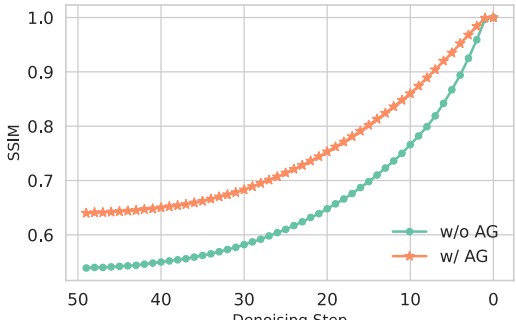

Figure 15: **Quantitatively analysis on the clustering effect of TFSA.** We calculate the SSIM between noised latents $z_t$ ($1 \leq t \leq 49$) and their corresponding clean latent $z_0$.

in Fig. 9); (**ii**) why attention guidance enhances the details and colors of the image (as shown in Fig. 6 and 8); and (**iii**) why applying attention guidance in the later stages of denoising does not enhance the image details and colors (as shown in Fig. 10).

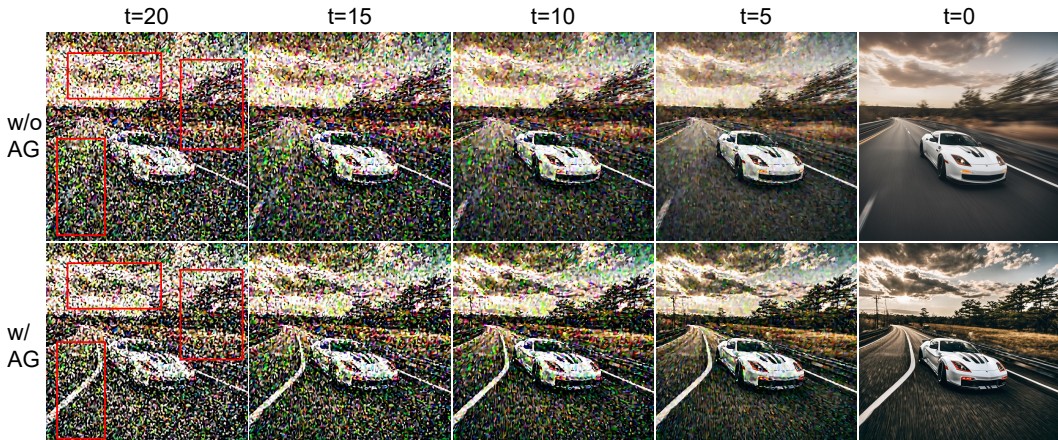

Figure 14: **Denoising visualization for the ablation of attention guidance.** As indicated by the red boxes, the clustering effect of TFSA prompts earlier structural emergence, delivering better prior for subsequent fine-detail generation. Resolution: $1024 \times 1024$.

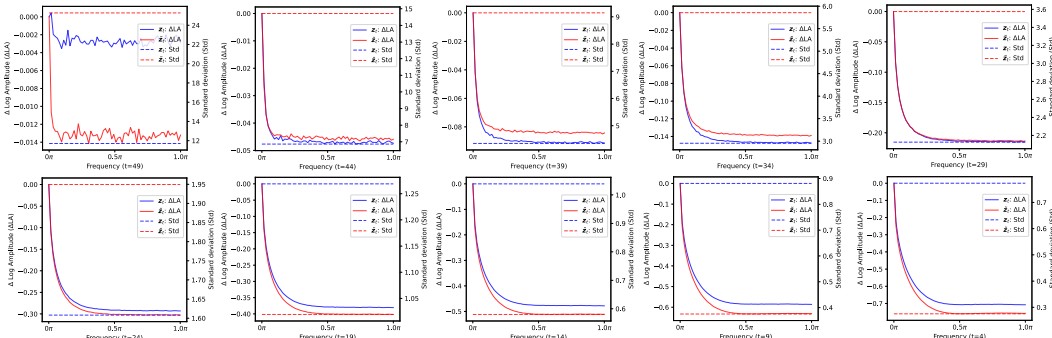

Figure 16: **The Fourier transform of the latent representation and the mean of the standard deviations across all channels.** $z_t$ is represented in blue, while $\tilde{z}_t$ is represented in red; the Fourier transforms are shown as solid lines, and the standard deviations are shown as dashed lines. The results are based on the generation process of 5k images.

To explain the aforementioned three points, as shown in Fig. 16, we calculate the Fourier transforms of $z_t$ (blue solid line) and $\tilde{z}_t$ (red solid line), along with the mean of the standard deviations for all their channels (dashed line). It can be observed that TFSA significantly alters the relative amplitudes of the high- and low-frequency components in the latent representations during the initial denoising steps (from $t = 49$ to $t = 47$), particularly affecting the low-frequency components, which results in structural deformation. During the early and middle stages of denoising (from $t = 44$ to $t = 29$), TFSA increases the amplitudes of high-frequency components in the latent representations, which explains why attention guidance leads to richer details and colors. In the later stages of denoising (from $t = 28$ to $t = 0$), TFSA slightly suppresses the high-frequency components of the latent representations while almost leaving the low-frequency components unchanged. This explains why applying attention guidance in the later stages of denoising cannot enrich details and colors of the generated images.

Additionally, Fig. 16 shows that TFSA increases the standard deviation of $\tilde{z}_t$ during the early and middle stages of denoising, while decreasing it in the later stages. The trend of the standard deviation changes is closely consistent with the variation in the amplitude of the high-frequency components. We conjecture that this is because the amount of information in the latent representations is positively correlated with the standard deviation, where a larger standard deviation corresponds to more image details and larger high-frequency components.

### A.3 Visualization of Attention Maps in TFSA

To further demonstrate the clustering effect of TFSA on related tokens, we visualize its attention maps. As shown in Fig. 17, without using projection matrices, the correlations between tokens are determined jointly by their represented colors and semantics. For example, in Fig. 17(a), the key

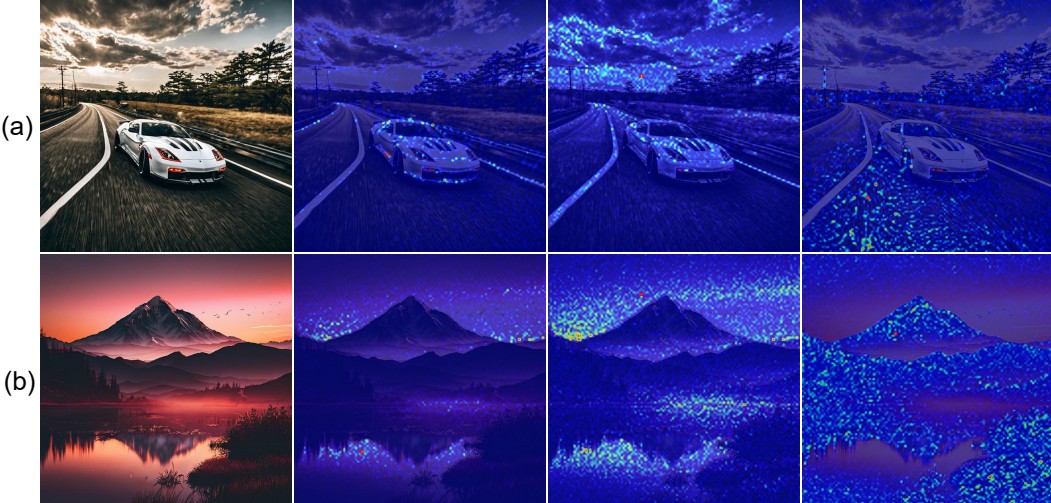

Figure 17: **Visualization of attention maps in TFSA.** The query tokens are highlighted with red boxes, and the heatmap color ranges from blue to red, indicating increasing correlation strength between the key tokens and the query tokens. Resolution: $1024 \times 1024$. Zoom-in for a better view.

tokens correlated with the query token at the selected car location are related not only to the car itself (*i.e.*, the concept of the car) but also to its color. TFSA leverages such correlations to fuse token information, thereby accelerating the formation of the overall image layout.

## B Supplementary Qualitative Comparison of §4.3

Fig. 18 presents additional qualitative comparison results. MultiDiffusion continues to struggle with maintaining global consistency; as indicated by the red boxes, DemoFusion tends to produce repetitive content, a problem somewhat alleviated in AccDiffusion but not fully resolved. As highlighted by the black boxes, another issue with AccDiffusion is the presence of noticeable streak artifacts in the images.

## C Supplementary Ablation Experiments of §5

### C.1 Further Qualitative Analysis of Attention Guidance

Fig. 19 provides additional qualitative ablation results on attention guidance. Individual preferences for contrast, color vividness, and detail richness may vary. attention guidance allows users to adjust parameters such as the guidance scale to synthesize images according to their preferences.

### C.2 Ablation on the hyper-parameters of Attention Guidance

**Quantitative analysis of guidance scale.** We sampled 1k prompts, fixed $\eta_1 = 0.06, \eta_2 = [0.2]$ and performed ablation studies for guidance scale $\gamma$. The quantitative results are shown in Table 5. Considering all metrics, we find that $\gamma = 0.004$ achieved better quantitative results.

**Quantitative analysis of delay rate.** We sampled 1k prompts, fixed $\gamma = 0.004, \eta_2 = [0.2]$ and performed ablation studies for delay rate $\eta_1$. Table 6 presents the experimental results, indicating that better results can be achieved when $\eta_1 = 0.06$. This means that appropriately delaying the effect of attention guidance can further enhance the quality of the generated images.

### C.3 Ablation on Progressive Scheduler Value

This section presents the results of quantitative ablation analysis on the progressive scheduler $\eta_2$ in the second stage of RepLDM. We fixed $\gamma = 0, \eta_1 = 0$, sampled 500 prompts, and generated 1k images to investigate the optimal value of the progressive scheduler. Table 7 presents the quantitative results, indicating that using an excessively large progressive scheduler may lead to a decline in image quality.

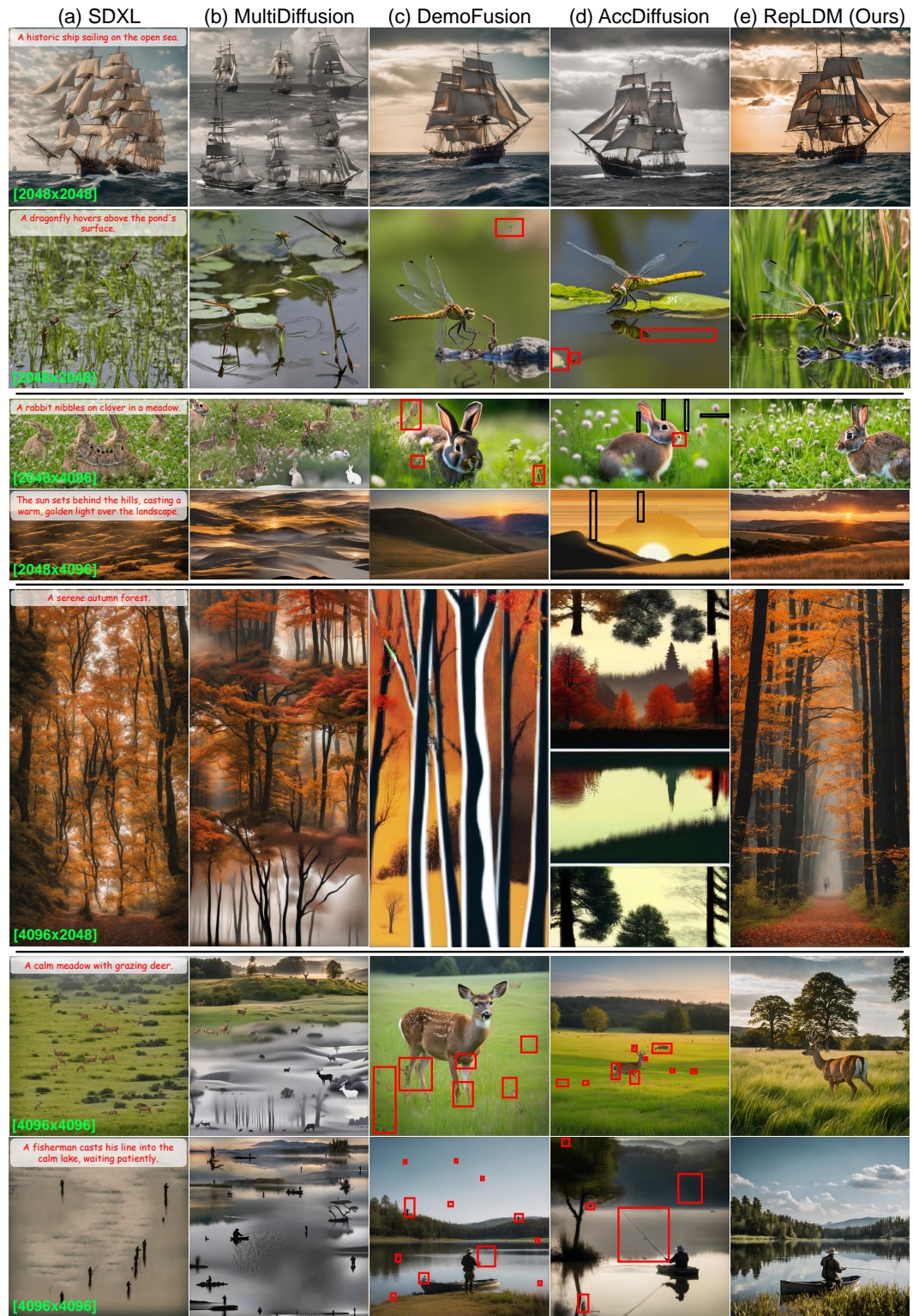

Figure 18: **Qualitative comparison with other baselines.** Zoom-in for a better view.

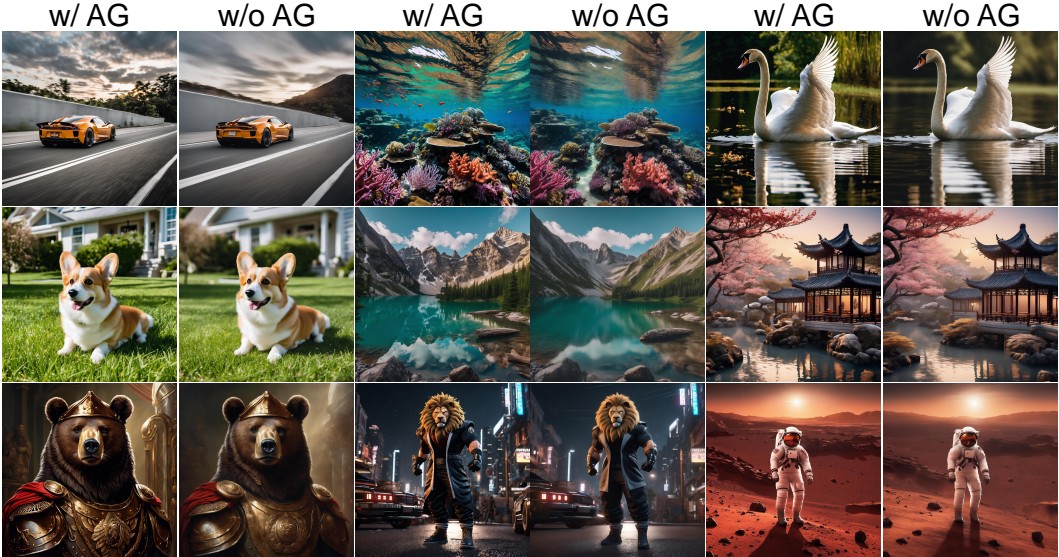

| w/ AG | w/o AG | w/ AG | w/o AG | w/ AG | w/o AG |

Figure 19: **Further qualitative analysis of attention guidance (AG).** Using attention guidance significantly enhances image quality. The details were enriched, for example: the clouds in the sky, ripples on the water, reflections on the lake, and even the expressions in a person's eyes. Resolution: 2048×2048. Best viewed **ZOOMED-IN**.

Table 5: **Quantitative ablation experiments on the guidance scale $\gamma$.** The best results are marked in **bold**, and the second best results are marked by underline.

| Method | 1024 × 1024 | | | | | 1600 × 1600 | | | | | 2048 × 2048 | | | | |
|---|---|---|---|---|---|---|---|---|---|---|---|---|---|---|---|
| | FID ↓ | IS ↑ | FID$_c$ ↓ | IS$_c$ ↑ | CLIP ↑ | FID ↓ | IS ↑ | FID$_c$ ↓ | IS$_c$ ↑ | CLIP ↑ | FID ↓ | IS ↑ | FID$_c$ ↓ | IS$_c$ ↑ | CLIP ↑ |
| $\gamma = 0.000$ | 90.85 | 58.18 | 21.21 | **17.69** | 25.09 | 90.91 | 54.74 | **21.45** | 15.41 | 24.93 | 91.78 | 59.08 | 21.57 | 17.36 | 24.86 |
| $\gamma = 0.001$ | 90.50 | 58.04 | **21.34** | 16.76 | 25.08 | 91.17 | 54.31 | 21.19 | 15.47 | 24.93 | 91.40 | 58.75 | **21.87** | 15.85 | 24.86 |
| $\gamma = 0.002$ | 89.82 | 57.54 | 21.28 | 17.04 | 25.08 | 90.39 | **53.71** | 21.26 | 15.00 | 24.97 | 90.81 | 58.34 | 21.45 | 17.16 | 24.90 |
| $\gamma = 0.003$ | 90.10 | 57.08 | 20.80 | 16.61 | 25.08 | 90.56 | 53.95 | 21.35 | 15.46 | 24.98 | 90.87 | 58.40 | 21.47 | **17.60** | 24.92 |
| $\gamma = 0.004$ | **89.40** | 56.64 | 20.96 | 16.63 | 25.09 | **89.91** | 54.23 | 20.91 | **15.54** | 25.01 | 90.11 | 58.11 | 21.18 | 16.78 | 24.94s |
| $\gamma = 0.005$ | 90.17 | 57.50 | 20.89 | 16.34 | 25.12 | 90.24 | 55.19 | 20.67 | 15.21 | 25.02 | 90.46 | 58.91 | 20.79 | 16.87 | 24.97 |
| $\gamma = 0.006$ | 89.79 | 58.18 | 20.33 | 15.93 | 25.16 | 90.36 | 56.71 | 20.33 | 14.59 | 25.06 | 90.32 | 59.86 | 20.37 | 16.12 | 25.00 |
| $\gamma = 0.007$ | 90.42 | 60.29 | 20.07 | 16.20 | 25.21 | 90.91 | 59.35 | 20.36 | 14.16 | 25.12 | 90.86 | 61.81 | 20.14 | 15.70 | 25.06 |
| $\gamma = 0.008$ | 91.64 | 63.63 | 19.66 | 14.25 | **25.25** | 91.98 | 63.93 | 19.13 | 13.71 | 25.13 | 92.16 | 64.82 | 19.59 | 14.24 | 25.08 |
| $\gamma = 0.009$ | 94.29 | 67.87 | 19.15 | 13.00 | 25.25 | 94.38 | 70.21 | 19.45 | 12.12 | **25.16** | 94.39 | 68.84 | 19.22 | 13.63 | **25.12** |

Table 6: **Quantitative ablation experiments on the delay rate $\eta_1$.** The best results are marked in **bold**, and the second best results are marked by underline.

| Method | 1024 × 1024 | | | | | 1600 × 1600 | | | | | 2048 × 2048 | | | | |
|---|---|---|---|---|---|---|---|---|---|---|---|---|---|---|---|
| | FID ↓ | IS ↑ | FID$_c$ ↓ | IS$_c$ ↑ | CLIP ↑ | FID ↓ | IS ↑ | FID$_c$ ↓ | IS$_c$ ↑ | CLIP ↑ | FID ↓ | IS ↑ | FID$_c$ ↓ | IS$_c$ ↑ | CLIP ↑ |
| $\eta_1 = 0.00$ | 89.98 | 58.29 | 20.74 | 16.48 | 25.06 | 90.89 | 55.54 | 21.00 | 14.42 | 24.98 | 90.75 | 59.41 | 20.54 | 16.99 | 24.91 |
| $\eta_1 = 0.02$ | 89.96 | 57.67 | 20.99 | **16.87** | 25.05 | 90.76 | 54.77 | 21.08 | 15.35 | 24.95 | 91.78 | 59.08 | **21.57** | **18.16** | 24.86 |
| $\eta_1 = 0.04$ | 89.47 | 57.28 | 20.98 | 16.63 | 25.07 | 90.22 | 54.14 | 20.86 | 15.43 | 24.98 | 90.52 | 58.47 | 20.76 | 17.02 | 24.91 |
| $\eta_1 = 0.06$ | 89.44 | 56.64 | 20.92 | 16.58 | **25.11** | 89.91 | 54.23 | 20.91 | 15.54 | 25.01 | 90.11 | 58.11 | 21.18 | 16.78 | 24.94 |
| $\eta_1 = 0.08$ | 89.95 | 56.97 | 21.05 | 16.76 | 25.09 | **89.87** | 54.10 | 21.22 | 15.65 | 24.98 | 90.74 | 58.45 | 20.99 | 17.06 | **24.92** |
| $\eta_1 = 0.10$ | **89.29** | 56.88 | **21.11** | 16.84 | 25.09 | 89.97 | 53.99 | 21.04 | 15.37 | 24.99 | 90.41 | 58.45 | 20.99 | 17.12 | 24.92 |
| $\eta_1 = 0.12$ | 89.84 | 57.32 | 21.05 | 16.58 | 25.08 | 90.00 | 53.85 | 21.24 | **15.81** | 24.93 | 90.24 | 58.45 | 21.24 | 17.36 | 24.90 |
| $\eta_1 = 0.14$ | 89.85 | 57.12 | 20.91 | 16.40 | 25.09 | 90.06 | 53.83 | 21.33 | 15.62 | **24.99** | 90.69 | 58.25 | 21.17 | 16.74 | 24.91 |
| $\eta_1 = 0.16$ | 90.06 | 57.28 | 21.10 | 16.53 | 25.09 | 90.91 | 54.74 | **21.45** | 15.41 | 24.93 | 90.76 | 58.37 | 20.97 | 16.87 | 24.91 |
| $\eta_1 = 0.18$ | 90.16 | 57.29 | 20.88 | 15.10 | 25.08 | 90.26 | **53.79** | 21.06 | 15.07 | 24.97 | 90.78 | 58.33 | 21.05 | 17.21 | 24.90 |

# D  Ablation on the Attention Guidance Components

## D.1  Ablation on the Guidance Scale Decay Strategy

To investigate the impact of different guidance scale decay strategies, we conduct ablation studies using two additional schemes—linear decay and exponential decay—and analyze their quantitative and qualitative performance. For quantitative ablation, we generate 2k samples at a resolution of $2048 \times 2048$ using each strategy and calculate the criterions on the SAM benchmark. Table 8 shows that different strategies yield similar results, indicating that RepLDM is not sensitive to a specific decay strategy. Fig. 20 illustrates the qualitative results. Qualitatively, these decay strategies also produce similar visual experience.

Table 7: **Quantitative ablation study of the progressive scheduler Value**. The best results are marked in **bold**, and the second best results are marked by underline.

| Method | 1600 × 1600 | | | | | 2048 × 2048 | | | | |
|---|---|---|---|---|---|---|---|---|---|---|
| | FID ↓ | IS ↑ | FID$_c$ ↓ | IS$_c$ ↓ | CLIP ↑ | FID ↓ | IS ↑ | FID$_c$ ↓ | IS$_c$ ↑ | CLIP ↑ |
| SDXL | 101.56 | 25.78 | 73.67 | 21.23 | 26.87 | 112.64 | 18.44 | 79.03 | 20.61 | 26.55 |
| $\eta_2 = [0.9]$ | 94.59 | 27.04 | 67.60 | 23.01 | 26.97 | 97.14 | 24.48 | 64.34 | 22.14 | 26.59 |
| $\eta_2 = [0.8]$ | 93.13 | 28.80 | 65.67 | 24.83 | 26.99 | 93.93 | 26.75 | 60.84 | 23.27 | 26.77 |
| $\eta_2 = [0.7]$ | **92.05** | 29.44 | 65.35 | 24.97 | 27.07 | 92.50 | 28.17 | 57.34 | 24.05 | 26.93 |
| $\eta_2 = [0.6]$ | 92.94 | 30.79 | 64.57 | 24.29 | 27.11 | 91.86 | 30.45 | 55.38 | 24.96 | 26.98 |
| $\eta_2 = [0.5]$ | 92.73 | 30.65 | 63.43 | 24.26 | 27.13 | 91.80 | 31.18 | 54.32 | 24.48 | 27.02 |
| $\eta_2 = [0.4]$ | 93.04 | 30.96 | 63.33 | 24.77 | 27.14 | **91.71** | **32.47** | 53.72 | 25.16 | 27.03 |
| $\eta_2 = [0.3]$ | 92.93 | 30.91 | **63.09** | 24.84 | 27.15 | 92.39 | 30.72 | 53.32 | **26.63** | 27.07 |
| $\eta_2 = [0.2]$ | 93.09 | **31.17** | 63.23 | **25.71** | 27.17 | 92.71 | 30.45 | **53.19** | 26.19 | 27.12 |
| $\eta_2 = [0.1]$ | 93.44 | 30.69 | 63.75 | 25.18 | **27.22** | 92.94 | 30.69 | 53.77 | 24.71 | **27.18** |

Table 8: **Ablation on the guidance scale decay strategies.** The best results are marked in **bold**, and the second best results are marked by underline.

| Strategies | FID ↓ | IS$_c$ ↑ | FID$_c$ ↓ | IS$_c$ ↑ | CLIP ↑ |
|---|---|---|---|---|---|
| Linear | 66.2 | 21.5 | 47.2 | **20.3** | **25.4** |
| Exponential | 66.8 | **21.8** | **47.0** | 16.3 | 25.3 |
| Cosine (default) | **66.0** | 21.0 | 47.4 | 17.5 | 25.1 |

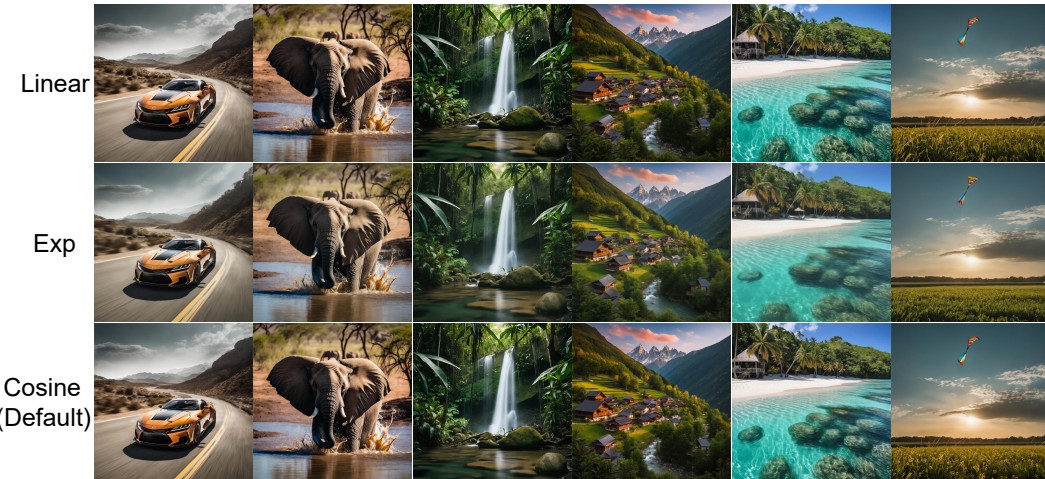

Figure 20: **Qualitative ablation on guidance scale decay ctrategies.**

## D.2 Ablation on the Attention Calculation Paradigm

For TFSA, our objective is to remove the learnable parameters from the Self-Attention mechanism, while maintaining its computational paradigm as unchanged as possible. In TFSA, $Q$, $K$, and $V$ are identical. Therefore, TFSA is a totally symmetric formula. As analyzed before, this paradigm encourages the clustering of semantically related tokens, and finally leads to finer details and richer colors. An interesting question arises: if we spatially downsample $Q$, $K$, or $V$ before applying TFSA and reformulate it into an asymmetric paradigm (denoted as TFSA-A), would TFSA-A encourage the model to attend more explicitly from fine details to coarse structures?

To answer this question, we design an asymmetric variants, TFSA-A. Specifically, TFSA-A performs a 2×2 pooling operation to downsample the $K$ and $V$ matrices before the attention calculation operation, ensuring that the output of $\mathrm{Softmax}(QK^T/\sqrt{d})V$ remains the of shape $(hw) \times c$. Table 9 shows that TFSA-A produces comparable quantitative results. In Fig. 21, we observe that although TFSA-A achieves quantitative results comparable to those of TFSA, its visual quality is significantly inferior. In fact, TFSA-A tends to reduce image details. This aligns with our hypothesis: the $2 \times 2$ pooling acts as a low-pass filter, causing the loss of fine-grained information in the latent representations and leading the model to focus more on low-frequency structures.

## E Further Model Efficiency Analysis

**Computational complexity analysis of TFSA.** Note that attention guidance is only applied during the first stage of generation. Assume we have a HR image $x_0$ with a resolution of $H \times W \times C$. we

Table 9: **Ablation on the attention calculation paradigm.** The best results are marked in **bold**, and the second best results are marked by underline.

| Paradigm | FID $\downarrow$ | $IS_c \uparrow$ | $FID_c \downarrow$ | $IS_c \uparrow$ | CLIP $\uparrow$ |
|---|---|---|---|---|---|
| w/o guidance | 66.8 | 21.6 | 47.5 | 17.4 | **25.3** |
| w/ TFSA-A | 67.4 | **22.6** | 47.9 | **20.4** | **25.3** |
| w/ TFSA | **66.0** | 21.0 | **47.4** | 17.5 | 25.1 |

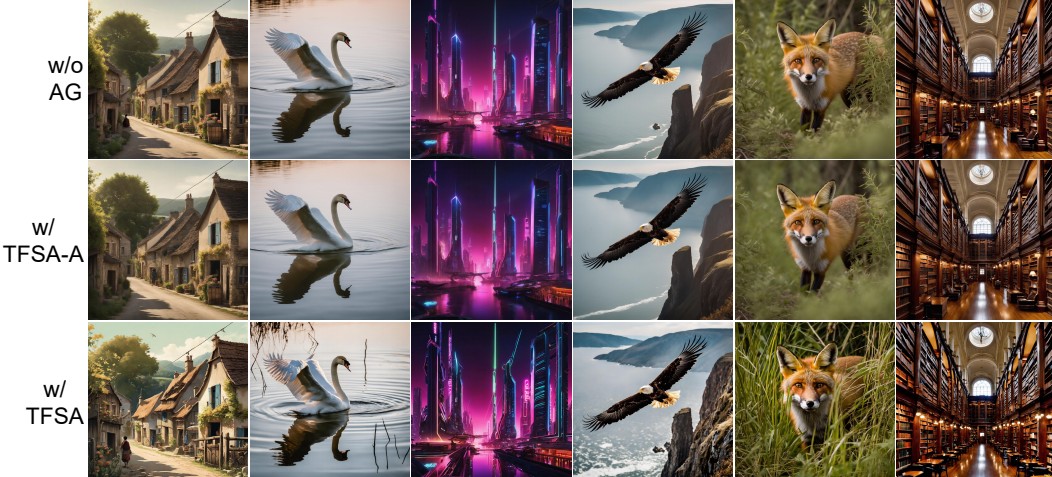

Figure 21: **Ablation on the attention calculation paradigm.** Resolution: $2048 \times 2048$.

encode the image $x_0$ into latent space and obtain latent representation $z_0 \in \mathbb{R}^{h \times w \times c}$. Before feeding $z_0$ into TFSA, we reshape it to a $(hw) \times c$ matrice. The computation of TFSA follows a formulation similar to that of self-attention: $\text{Softmax}(z_0 z_0^T / \sqrt{c})z$. Thus, the computational complexity of TFSA is $O((hw)^2 c)$. Taking SDXL as an example, the training resolution is $H = 1024$, $W = 1024$. After VAE encoding, $c = 4$, $h = H/8 = 128$, $w = W/8 = 128$. For each denoising step, the FLOPs of TFSA is approximately $2 \times (h \times w)^2 \times c$, which is around 2.15 GFLOPs—negligible compared to the FLOPs of the denoising network (several TFLOPs per step).

**How does pixel space upsampling accelerate generation?** To answer this question, we analyze the time consumption of each component in DemoFusion and RepLDM when generating images at the resolution of $4096 \times 4096$.

Table 10: **The time consumption of DemoFusion when generating** $4096 \times 4096$ **resolution images.**

| Metric | Denoise 1024 | Denoise 2048 | Denoise 3072 | Denoise 4096 | Decode 4096 | Total |
|---|---|---|---|---|---|---|
| number of steps | 50 | 50 | 50 | 50 | - | 200 |
| Time (s) | 12 | 185 | 480 | 901 | 106 | 1684 |

Table 11: **The time consumption of RepLDM when generating** $4096 \times 4096$ **resolution images.** The intermediate encoding/decoding operations are highlighted in underline.

| Metric | Denoise 1024 | Decode 1024 | Encode 3304 | Denoise 3304 | Decode 3304 | Encode 4096 | Denoise 4096 | Decode 4096 | Total |
|---|---|---|---|---|---|---|---|---|---|
| number of steps | 50 | - | - | 5 | - | - | 10 | - | 65 |
| Time (s) | 12 | 0 | 12 | 20 | 64 | 11 | 118 | 106 | 343 |

Table 10 shows that denoising at high resolutions is a time-consuming process. DemoFusion requires substantial generation time because it performs the full denoising process at high resolutions. Note that, compared with the cost of the denoising process at high resolutions, the costs of encoding and decoding are negligible. Table 11 shows that RepLDM significantly accelerates generation by substantially reducing the number of denoising steps at high resolutions. This is because RepLDM performs pixel space upsampling through multiple rounds of encoding and decoding, producing high-quality low-resolution images that serve as better initialization. As a result, RepLDM can significantly reduce the number of sampling steps required for HR generation, thereby accelerating the process. Moreover, Table 11 shows that the additional overhead from multiple intermediate encoding and decoding operations is also relatively minor compared to the total generation cost.

**Further efficiency comparison across different models.** To provide a more comprehensive assessment of model efficiency, we further report the NFE and FLOPs of different models when generating a single image at resolutions of $2048 \times 2048$ and $4096 \times 4096$. Tables 12 and 13 show

that RepLDM significantly reduces the NFE and FLOPs required for inference by decreasing the number of denoising steps at high resolutions, thereby substantially reducing the time needed to generate HR images.

Table 12: **Inference cost of generating a** $2048 \times 2048$ **Image for different models.**

| Model | SDXL [36] | MultiDiff. [1] | ScaleCrafter [14] | HiDiff. [56] | UG [21] | DemoFusion [6] | AccDiff. [28] | RepLDM |
|---|---|---|---|---|---|---|---|---|
| NFE | 50 | 50 | 50 | 50 | 80 | 100 | 100 | 60 |
| TFLOPs | 3010 | 5420 | 2437 | 1857 | 3608 | 9015 | 8597 | 1140 |
| Time (min) | 1.0 | 3.0 | 1.0 | 0.8 | 1.8 | 3.0 | 3.0 | 0.6 |

Table 13: **Inference cost of generating a** $4096 \times 4096$ **Image for different models.**

| Model | SDXL [36] | MultiDiff. [1] | ScaleCrafter [14] | HiDiff. [56] | UG [21] | DemoFusion [6] | AccDiff. [28] | RepLDM |
|---|---|---|---|---|---|---|---|---|
| NFE | 50 | 50 | 50 | 50 | 80 | 200 | 200 | 65 |
| TFLOPs | 12026 | 29566 | 9759 | 5211 | 12624 | 72167 | 74225 | 7140 |
| Time (min) | 8.0 | 15.0 | 19.0 | 3.4 | 11.1 | 25.0 | 26.0 | 5.7 |

**Qualitative analysis on the progressive upsampling stage.** To clearly illustrate the progressive upsampling process of RepLDM, we set $\eta_2 = [0.2, 0.2, 0.2]$ to generate $4096 \times 4096$ images. As shown in Fig. 22, the images generated at different sub-stages of RepLDM exhibit a high degree of consistency, with only minor differences in details. Since our task focuses on generating HR images rather than traditional image super-resolution, these differences in details are reasonable. As discussed in Table 10 and Table 11, for each denoising step, the time required for HR images is several times that for low-resolution images. Consequently, repeating a full denoising process at high resolution is extremely time-consuming [6, 28]. Considering that HR and low-resolution images should share the same low-frequency structure, and that DMs naturally generate low-frequency structures first during denoising [44, 53], RepLDM leverages the prior knowledge of low-frequency structures in low-resolution images, thereby effectively accelerating the generation process.

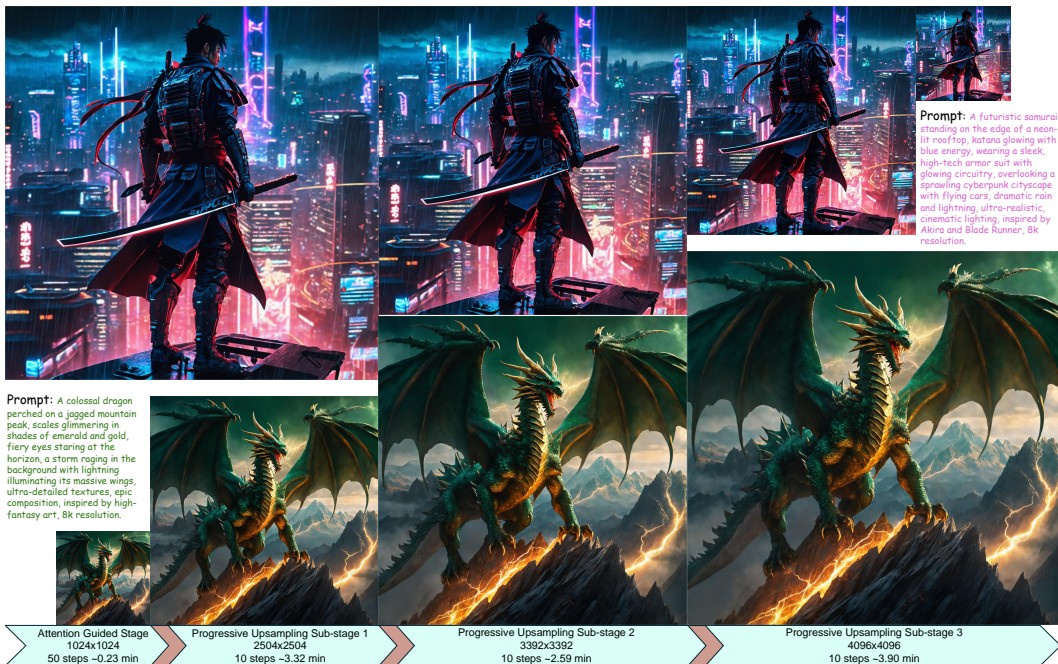

Figure 22: **Illustration of the progressive upsampling generation process.** The inference speed is evaluated on a single 3090 GPU.

# F  RepLDM Algorithm

The implementation details of RepLDM can be found in Algorithm 1, and further information is available in our code repository.

**Algorithm 1** RepLDM Inference Pipeline

---

**Require:** The number of inference time steps of the first stage $T_0$; progressive scheduler $\eta_2$; attention guidance scale $\gamma$; attention guidance delay rate $\eta_1$; the decay factor $\beta$; target image size tuple $(H', W')$; the denoising model $\mathcal{F}$; denoising model's training resolution tuple $(H, W)$; VAE encoder $\mathcal{E}$; VAE decoder $\mathcal{D}$; noise scheduler's hyper-parameter list $\bar{\alpha}_{1:T_0}$.

1: **Initialization:**
2: $\boldsymbol{z}_{T_0}^{(0)} = \boldsymbol{\epsilon} \sim \mathcal{N}(0, \boldsymbol{I})$ {Sampling from Standard Gaussian Distribution}
3: $n_{\text{stages}} = \text{length}(\eta_2) + 1$ {Get the total number of denoising stages}
4: $r' = \frac{H'}{W'}$ {Keep the aspect ratio and number of pixels unchanged}
5: $H^{(0)} = \text{ceil}(\sqrt{H \times W \times r'})$
6: $W^{(0)} = \text{ceil}(\sqrt{\frac{H \times W}{r'}})$
7: $H^{(n)} = H'$
8: $W^{(n)} = W'$
9: $area_{\text{list}} = \text{linspace}(H^{(0)} \times W^{(0)}, H^{(n)} \times W^{(n)}, n_{\text{stages}})$ {Upsampling according to the number of pixels}
10: $H_{\text{list}} = [\text{ceil}(\sqrt{i \times r'}) \quad \text{for} \quad i \quad \text{in} \quad area_{\text{list}}]$ {Get the height and width of each stage}
11: $W_{\text{list}} = [\text{ceil}(\sqrt{i/r'}) \quad \text{for} \quad i \quad \text{in} \quad area_{\text{list}}]$
12: $k_{\text{denoising}} = [T_0]$ {Get the number of denoising steps for each stage}
13: $k_{\text{denoising}}.\text{extend}([i \times T_0 \quad \text{for} \quad i \quad \text{in} \quad \eta_2])$
14: $k = T_0 \times \eta_1$ {Obtain the number of delay steps}
15: $\gamma_{\text{list}} = [\gamma(\frac{cos(\frac{T-k-i}{T-k}\pi)+1}{2})^\beta \quad \text{for} \quad i = 1, ..., T-k]$ {Obtain the guidance scale for each step}
16: **Denoising:**
17: **for** $s = 0, \ldots, n_{\text{stages}} - 1$ **do**
18: $\quad n_{\text{steps}} \leftarrow k_{\text{denoising}}[s]$
19: $\quad$ **if** $s \geq 1$ **then**
20: $\quad\quad \boldsymbol{x}^{(s)} \leftarrow \text{upsample}(\boldsymbol{x}^{(s-1)}, H_{\text{list}}[s], W_{\text{list}}[s])$ {Upsampling in pixel space}
21: $\quad\quad \boldsymbol{z}_0^{(s)} \leftarrow \mathcal{E}(\boldsymbol{x}^{(s)})$
22: $\quad\quad \boldsymbol{z}_{n_{\text{steps}}}^{(s)} \sim \mathcal{N}(\sqrt{\bar{\alpha}[n_{\text{steps}}]}\boldsymbol{z}_0^{(s)}, (1 - \bar{\alpha}[n_{\text{steps}}])\boldsymbol{I})$
23: $\quad$ **end if**
24: $\quad$ **for** $t = n_{\text{steps}} - 1, \ldots, 0$ **do**
25: $\quad\quad \boldsymbol{z}_t^{(s)} \leftarrow \mathcal{F}(\boldsymbol{z}_{t+1}^{(s)}, t+1)$ {Denoising}
26: $\quad\quad$ **if** $s == 0$ and $t \leq T - 1 - k$ **then**
27: $\quad\quad\quad \boldsymbol{z}_t^{(s)} \leftarrow \gamma_{\text{list}}[t]\text{PFSA}(\boldsymbol{z}_t^{(s)}) + (1 - \gamma_{\text{list}}[t])\boldsymbol{z}_t^{(s)}$ {Attention Guidance}
28: $\quad\quad$ **end if**
29: $\quad$ **end for**
30: $\quad \boldsymbol{x}^{(s)} \leftarrow \mathcal{D}(\boldsymbol{z}_0^{(s)})$ {Obtain the pixel space image}
31: **end for**

---

# G Robustness Analysis

In this section, we conduct a robustness analysis to complement the experiments in §4.2, providing a more comprehensive evaluation of the models' performance. Our robustness analysis is conducted from two perspectives: (**i**) we vary the random seeds and repeat each experiment three times to compute the mean and standard deviation of all results; (**ii**) we randomly sample 20k HR images from the HR subset of LAION-5B dataset [41] to construct a new benchmark for evaluating the models' generalization performance. Since HR generation requires substantial computational resources, we analyze the four best-performing models from Table 1, *i.e.*, HiDiffusion, DemoFusion, AccDiffusion, and RepLDM.

**Analysis on the SAM benchmark.** We maintain the exact experimental settings as in §4.2 and conduct the analysis at resolutions of $2048 \times 2048$ and $4096 \times 4096$. Table 14 shows that RepLDM continues to exhibit superior performance across the repeated experiments.

Table 14: **Robustness analysis on the SAM benchmark**. The best results are marked in **bold**.

| Method | 2048 × 2048 | | | | | 4096 × 4096 | | | | |
|---|---|---|---|---|---|---|---|---|---|---|
| | FID ↓ | IS ↑ | FID$_c$ ↓ | IS$_c$ ↑ | CLIP ↑ | FID ↓ | IS ↑ | FID$_c$ ↓ | IS$_c$ ↑ | CLIP ↑ |
| HiDiff. [56] | 80.29±0.57 | 17.18±0.40 | 63.55±0.63 | 15.26±0.76 | 24.95±0.04 | 144.24±0.84 | 12.71±0.14 | 146.62±0.32 | 7.48±0.28 | 21.18±0.05 |
| DemoF. [6] | 71.89±0.60 | 22.10±0.37 | 53.58±0.22 | 19.21±0.27 | 25.21±0.01 | 101.83±0.49 | 20.81±0.11 | 63.60±0.46 | 14.92±1.24 | **24.75**±0.03 |
| AccDiff. [28] | 71.37±0.48 | 21.21±0.32 | 53.04±0.33 | 19.24±1.72 | 25.13±0.01 | 102.41±1.40 | 19.88±0.24 | 65.86±0.17 | 12.73±0.71 | 24.65±0.02 |
| RepLDM | **66.08**±0.02 | **22.13**±0.74 | **47.31**±0.11 | **20.38**±2.03 | **25.30**±0.12 | **91.46**±0.61 | **21.63**±0.46 | **58.93**±0.20 | **15.02**±0.16 | 24.62±0.02 |

**Analysis on the LAION-5B benchmark.** Considering that only 1K samples were used for the $4096 \times 4096$ resolution in §4.2, which may lead to unstable metric evaluations, we double the number of samples to 2k for this resolution in the current experiment. Regarding evaluation metrics, since IS may lead to high variances beyond ImageNet, we follow some recent studies and adopt Kernel Inception distance (KID) for more accurate evaluation [20, 37]. Table 15 shows that on the LAION benchmark, RepLDM still demonstrates superior performance, surpassing previous SOTA models across all metrics.

Table 15: **Robustness analysis on the LAION-5B benchmark**. The best results are marked in **bold**. Since the magnitude of KID is relatively small, we multiply its mean and standard deviation by $10^3$.

| Method | $2048 \times 2048$ | | | | | $4096 \times 4096$ | | | | |
|---|---|---|---|---|---|---|---|---|---|---|
| | FID $\downarrow$ | KID $\downarrow$ | FID$_c \downarrow$ | KID$_c \downarrow$ | CLIP $\uparrow$ | FID $\downarrow$ | KID $\downarrow$ | FID$_c \downarrow$ | KID$_c \downarrow$ | CLIP $\uparrow$ |
| HiDiff. [56] | 48.17±0.41 | 8.06±0.20 | 36.26±0.37 | 10.93±0.11 | 23.16±0.03 | 92.81±0.78 | 35.36±0.60 | 120.26±0.91 | 103.45±0.27 | 18.55±0.06 |
| DemoF. [6] | 34.15±0.31 | 4.50±0.05 | 21.38±0.17 | 6.80±0.06 | 25.44±0.02 | 37.03±0.27 | 5.71±0.14 | 30.77±0.36 | 16.12±0.22 | 25.12±0.04 |
| AccDiff. [28] | 34.49±0.31 | 4.92±0.08 | 22.71±0.17 | 8.57±0.11 | 24.90±0.02 | 38.56±0.23 | 7.21±0.20 | 38.85±0.29 | 20.87±0.20 | 24.46±0.01 |
| RepLDM | **34.08**±0.25 | **4.18**±0.04 | **20.30**±0.30 | **4.87**±0.13 | **25.78**±0.03 | **34.01**±0.26 | **4.13**±0.05 | **23.08**±0.26 | **12.08**±0.13 | **25.88**±0.04 |

# H Comparative and Ablation Analysis Based on StableDiffusion 2.1

## H.1 Comparison Experiments

To validate the generalization capability of RepLDM, we conducted extensive quantitative and qualitative analyses using StableDiffusion 2.1 (SD2.1) as the pretrained base model.

**Qualitative comparison.** Fig. 23 presents the results of the qualitative comparison. It can be observed that, when generating high-resolution images, SD2.1 also faces issues with repetitive object structures. ScaleCrafter often exhibits structural collapse during denoising with SD2.1, resulting in suboptimal performance. In contrast, RepLDM consistently produces high-quality results across all resolutions, highlighting the generalizability of the RepLDM generation framework.

**Quantitative comparison.** Since the code for using SD2.1 as the pretrained model in AccDiffusion and DemoFusion is not publicly available, we compare RepLDM with ScaleCrafter. We compared the model performance at four resolutions: $1536 \times 1536$, $1024 \times 2048$, $2048 \times 1024$, and $2048 \times 2048$. Considering that SD2.1's generation capabilities are weaker than SDXL, we set $\eta_2 = [0.2, 0.2, 0.3]$ for the experiments in this section, while keeping other settings consistent with §4.

Table 16 presents the results of the quantitative comparison, showing that RepLDM maintains strong performance when using SD2.1 as the pre-trained model. In contrast, ScaleCrafter performs suboptimally, as it tends to produce structural collapse in the generated images, a phenomenon that is more apparent in the qualitative analysis.

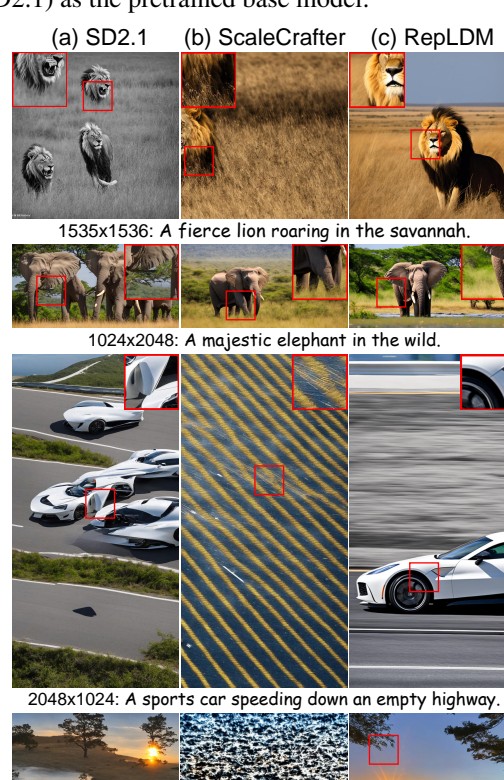

(a) SD2.1    (b) ScaleCrafter    (c) RepLDM

1535x1536: A fierce lion roaring in the savannah.

1024x2048: A majestic elephant in the wild.

2048x1024: A sports car speeding down an empty highway.

2048x2048: The sun rises, casting a warm glow over the land.

Figure 23: **Qualitative comparison using SD2.1 as the pretrained model**.

## H.2 Ablation Study on Attention Guidance

**Quantitative ablation.** Table 17 shows the results of the quantitative ablation on attention guidance using SD2.1 as the pretrained model. It can be observed that attention guidance leads to improvements in metrics. These improvements are more evident in the qualitative ablation analysis.

Table 16: **Quantitative comparison results based on SD2.1**. The best results are marked in **bold**.

| Method | 1536 × 1536 | | | | | 1024 × 2048 | | | | | 2048 × 1024 | | | | | 2048 × 2048 | | | | |
|---|---|---|---|---|---|---|---|---|---|---|---|---|---|---|---|---|---|---|---|---|
| | FID | IS | $FID_c$ | $IS_c$ | CLIP | FID | IS | $FID_c$ | $IS_c$ | CLIP | FID | IS | $FID_c$ | $IS_c$ | CLIP | FID | IS | $FID_c$ | $IS_c$ | CLIP |
| SD2.1 [39] | 95.4 | 17.8 | 83.4 | 15.8 | 25.0 | 85.8 | 15.9 | 76.1 | 16.3 | 25.2 | 101.8 | 15.8 | 79.8 | 16.8 | 24.6 | 121.7 | 14.4 | 92.7 | 14.4 | 24.5 |
| ScaleCrafter [14] | 140.4 | 10.6 | 136.4 | 9.7 | 21.9 | 150.0 | 10.1 | 139.3 | 10.1 | 21.7 | 149.8 | 10.4 | 135.6 | 11.5 | 21.8 | 144.2 | 10.4 | 135.2 | 10.3 | 23.4 |
| RepLDM | **60.3** | **21.0** | **50.6** | **18.3** | **25.4** | **61.1** | **19.9** | **54.1** | **18.4** | **25.0** | **63.7** | **19.2** | **50.4** | **18.2** | **24.7** | **60.5** | **21.5** | **48.8** | **17.2** | **25.3** |

Table 17: **Quantitative ablation results based on SD2.1**. The best results are marked in **bold**.

| Method | 1536 × 1536 | | | | | 1024 × 2048 | | | | | 2048 × 1024 | | | | | 2048 × 2048 | | | | |
|---|---|---|---|---|---|---|---|---|---|---|---|---|---|---|---|---|---|---|---|---|
| | FID | IS | $FID_c$ | $IS_c$ | CLIP | FID | IS | $FID_c$ | $IS_c$ | CLIP | FID | IS | $FID_c$ | $IS_c$ | CLIP | FID | IS | $FID_c$ | $IS_c$ | CLIP |
| w/o AG | 61.2 | 20.9 | **50.2** | **18.9** | 25.2 | 61.5 | 19.6 | **54.0** | **19.5** | 24.9 | 64.6 | **19.6** | **49.2** | 17.0 | 24.6 | 61.1 | 21.2 | **46.5** | **18.2** | 25.2 |
| w/ AG | **60.3** | **21.0** | 50.6 | 18.3 | **25.4** | **61.1** | **19.9** | 54.1 | 18.4 | **25.0** | **63.7** | 19.2 | 50.4 | **18.2** | **24.7** | **60.5** | **21.5** | 48.8 | 17.2 | **25.3** |

**Qualitative ablation.** Fig. 24 presents the ablation analysis of attention guidance based on SD2.1. From the figure, it can be observed that attention guidance also enhances detail richness and color vibrancy when using SD2.1, further demonstrating its generalization capability.

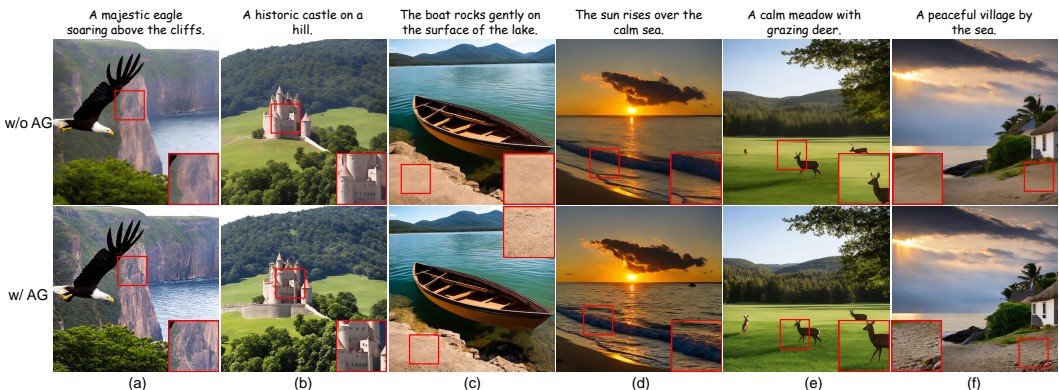

Figure 24: **Ablation study of attention guidance using SD2.1 as the pre-trained model**. Resolution: 2048 × 2048.

# I   Attention Guidance Also Works in Other Generation Frameworks

In this section, we apply attention guidance to other generative frameworks to demonstrate its generalization capability. Specifically, we apply attention guidance to the generative frameworks of HiDiffusion and DemoFusion, and perform both quantitative and qualitative ablation studies.

## I.1   Quantitative Ablation

In this section, considering the long inference time of DemoFusion, we perform quantitative ablation studies on attention guidance using the HiDiffusion generation frameworks at a resolution of 2048 × 2048. All experimental settings are consistent with those in §4.

Table 18 presents the quantitative ablation results using the HiDiffusion framework. It is evident that incorporating attention guidance improves HiDiffusion across all metrics. This is further corroborated by the qualitative analysis in Fig. 25, which demonstrates that attention guidance alleviates some of the structural collapses observed in HiDiffusion.

Table 18: **Quantitative ablation of attention guidance using HiDiffusion frameworks**. The best results are marked in bold.

| Method | FID | IS | $FID_c$ | $IS_c$ | CLIP |
|---|---|---|---|---|---|
| HiDiffusion [56] | 81.0 | 16.8 | 64.1 | 14.2 | **24.9** |
| HiDiffusion+AG | **79.4** | **17.0** | **62.4** | **14.6** | **24.9** |

## I.2   Qualitative Ablation

**HiDiffusion+attention guidance.** We incorporate attention guidance into the generative framework of HiDiffusion. Fig. 25 (a)-(c) demonstrate that using attention guidance effectively mitigates the issue of structural collapse in synthesized images. Fig. 25 (d)-(f) further show that attention guidance can also address the structural deformation inherent to HiDiffusion, enhance image details, and improve overall image quality.

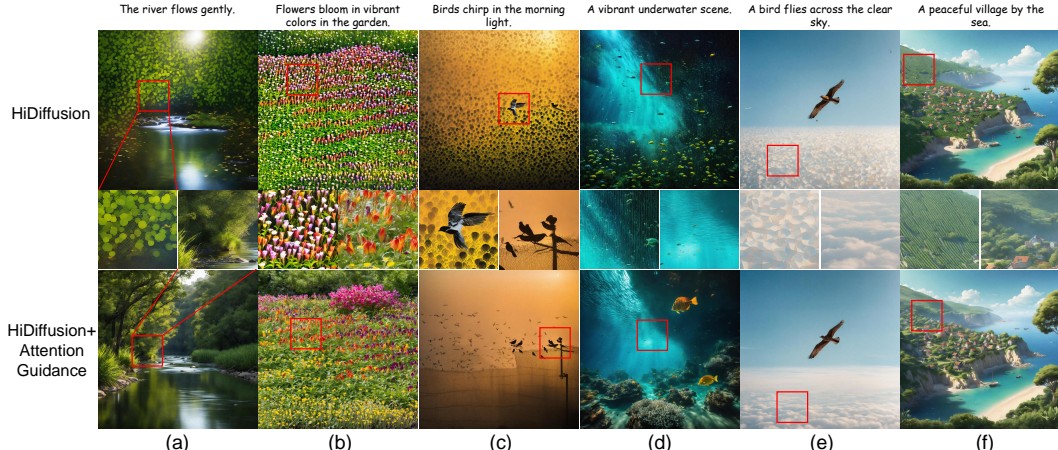

Figure 25: **Qualitative ablation of attention guidance in the HiDiffusion Framework**. All images have a resolution of $2048 \times 2048$. Figures (a)-(c) demonstrate that attention guidance can mitigate the issue of structural collapse in generated images, while Figures (d)-(f) show that attention guidance resolves structural deformation issues and enhances image details.

**DemoFusion+attention guidance.** We incorporate attention guidance into the generative framework of DemoFusion. As shown in Fig. 26 (a)-(c), attention guidance effectively mitigates the issue of repetitive structures in DemoFusion. Fig. 26 (d)-(f) further illustrate role of attention guidance in enriching image details and enhancing overall image quality.

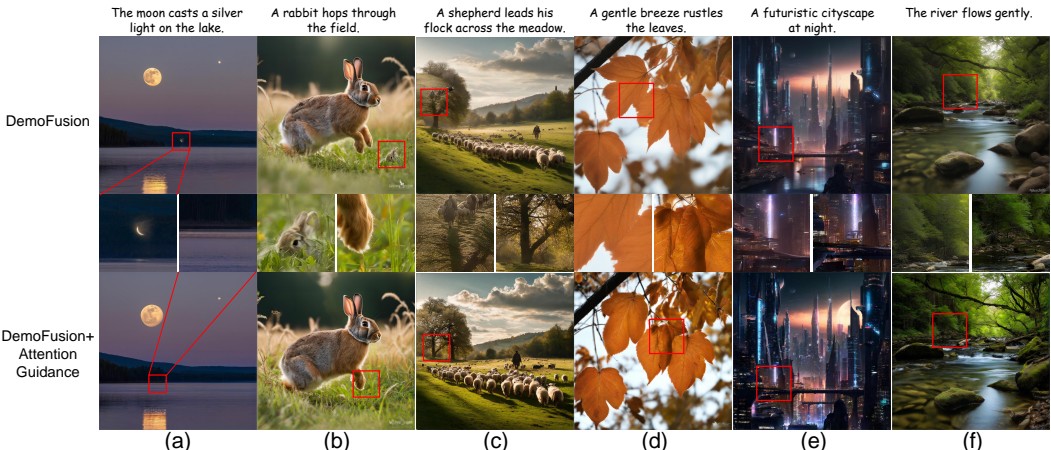

Figure 26: **Qualitative ablation of attention guidance in the DemoFusion Framework**. All images have a resolution of $2048 \times 2048$. Figures (a)-(c) demonstrate that attention guidance effectively mitigates the issue of repetitive structures in images, while Figures (d)-(f) showcase attention guidance's ability to enrich image details.

## J  Super-Resolved Images Tend to Lack High-Resolution Details

To explain why using super-resolution models to obtain HR images is sub-optimal, in this section, we conduct both qualitative and quantitative comparisons between RepLDM and the super-resolution results. Specifically, we use BSRGAN [54] to upsample the generated results of SDXL [36] at its training resolution.

**Quantitative results.** As shown in table. 19, the super-resolution model (SDXL + BSRGAN) demonstrate comparable performance in quantitative experiments, a phenomenon also observed in the DemoFusion's experiments. This is because super-resolution models can at least preserve the low-frequency structures of images without significant errors. However, quantitative metrics such as FID and IS, are widely recognized as insufficient for comprehensively evaluating the performance of model's generation. As a result, user studies are commonly employed to provide human-level

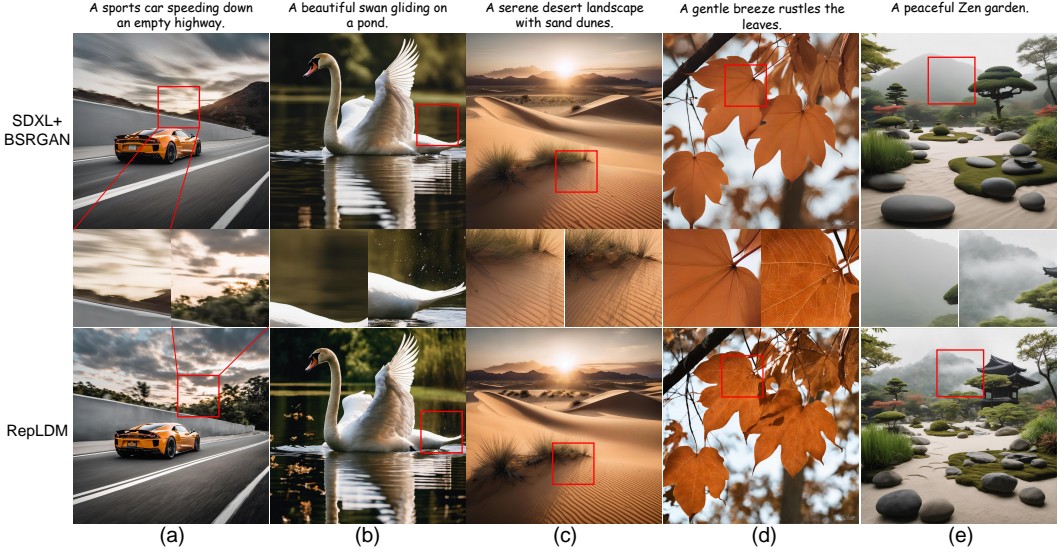

| A sports car speeding down an empty highway. | A beautiful swan gliding on a pond. | A serene desert landscape with sand dunes. | A gentle breeze rustles the leaves. | A peaceful Zen garden. |

(a) (b) (c) (d) (e)

Figure 27: **Qualitative comparison with SDXL+BSRGAN**. The prompts for the generated images are provided above the figures. The resolution of (a) to (c) are $2048 \times 2048$, and the resolution of (d) and (e) are $4096 \times 4096$.

evaluation with more intuition [11, 14, 22, 30, 36, 39, 55]. For example, in ScaleCrafter [14], they conducted both quantitative and user study analyses in comparison with the SD+SR approach. Their results show that, although ScaleCrafter performs worse than SD+SR on quantitative metrics, users significantly prefer the textures and details generated by ScaleCrafter. One important reason is that the goal of the SR model is to produce images consistent with the input, which limits its performance in high-resolution generation – needing more detail for true high-resolution visuals beyond simple smoothing [6, 10, 14, 22, 27, 28, 57].

Table 19: **Quantitative comparison results between RepLDM and SDXL+BSRGAN**. The best results are marked in **bold**.

| Method | 2048 × 2048 | | | | | 2048 × 4096 | | | | | 4096 × 2048 | | | | | 4096 × 4096 | | | | |
|---|---|---|---|---|---|---|---|---|---|---|---|---|---|---|---|---|---|---|---|---|
| | FID | IS | $FID_c$ | $IS_c$ | CLIP | FID | IS | $FID_c$ | $IS_c$ | CLIP | FID | IS | $FID_c$ | $IS_c$ | CLIP | FID | IS | $FID_c$ | $IS_c$ | CLIP |
| SDXL+BSRGAN | 66.2 | 21.1 | 47.5 | 16.6 | **25.7** | **80.7** | 19.8 | **50.2** | 12.3 | **25.1** | **92.7** | 17.6 | 57.9 | 12.1 | **24.9** | **90.0** | 20.9 | **56.0** | 13.8 | **25.2** |
| RepLDM | **66.0** | 21.0 | **47.4** | 17.5 | 25.1 | 89.0 | **20.3** | 56.0 | **19.0** | 25.0 | 93.2 | **19.5** | **56.9** | **16.5** | 24.9 | 90.6 | **21.1** | 59.0 | **14.8** | 24.6 |

**Qualitative results.** As shown in Fig. 27, compared to RepLDM, SDXL+BSRGAN, while maintaining decent image structure, fails to generate the level of detail expected from HR images. The absence of these details sometimes leads to the model's inability to simulate realistic scenes. For example, in Fig. 27 (c), SDXL+BSRGAN fails to generate realistic shadows.

## K  Memory Usage Analysis

We compare the GPU memory usage required by the models. Specifically, we test the minimum GPU memory requirements during model inference based on the model's open-source code. Table 20 shows the resource consumption of different models when generating images at various resolutions.

Table 20: **Model Memory Usage (GB).** The best results are marked in **bold**, and the second best results are marked by underline.

| Resolutions | 2048 × 2048 | 2048 × 4096 | 4096 × 4096 |
|---|---|---|---|
| SDXL [36] | 15.9 | **16.1** | 16.6 |
| MultiDiff. [1] | 22.0 | 16.8 | 16.8 |
| ScaleCrafter [14] | 17.4 | 17.6 | 19.1 |
| UG [21] | 23.9 | 16.5 | 18.0 |
| DemoFusion [6] | **15.2** | 18.4 | 16.8 |
| AccDiff. [28] | 22.1 | 23.0 | 22.1 |
| HiDiff. [56] | 23.9 | 16.2 | **16.2** |
| RepLDM | 16.0 | 21.1 | 23.8 |

It is worth noting that for HR image generation tasks, the memory bottleneck lies in the encoding and decoding of the VAE rather than interpolating the image in pixel space. To address the challenges of

encoding and decoding HR images, researchers typically employ tiled encoders and tiled decoders. In this work, we also utilize a tiled-encoder and decoder when generating ultra-high-resolution images, allowing us to generate images with resolutions up to $4096 \times 7280$ or higher on a 24GB VRAM NVIDIA 3090 GPU (as shown in Fig. 1).

It is important to note that different models have undergone varying degrees of additional optimization in their official open-source implementations. Specifically, some open-source codes utilize existing optimization tools, such as accelerate [9] or Flash Attention [3], which provide additional advantages in terms of inference speed and memory usage performance. To ensure a fair comparison, in Table 20, we did not use such additional optimizations in the implementation of RepLDM.

