# OpenReview forum: "RepLDM: Reprogramming Pretrained Latent Diffusion Models for High-Quality, High-Efficiency, High-Resolution Image Generation"
_NeurIPS.cc/2025/Conference — NeurIPS 2025 spotlight_

### Official Review · Reviewer_urNj · 2025-06-06

**Clarity:** 3
**Significance:** 2
**Originality:** 3
**Rating:** 5
**Confidence:** 4

**Summary:**

This paper proposes a training-free image generation framework called RepLDM, which aims to generate high-resolution images while preserving global structural consistency. To achieve this goal, the authors introduce a parameter-free self-attention mechanism (PFSA) and a pixel-space-based Progressive Upsampling Stage. Both quantitative and qualitative comparisons, as well as ablation studies, demonstrate the effectiveness of the proposed method.

**Questions:**

The concerns are as described in the Weaknesses section. Due to the uncertainty regarding the effectiveness of PFSA, I am assigning a borderline score for now out of caution. If the authors can adequately address my concerns during the rebuttal phase, I would consider raising my score.

**Ethical Concerns:**

["NO or VERY MINOR ethics concerns only"]

**Final Justification:**

We primarily focus on the effectiveness of PFSA, as it shows promise as a general plug-and-play method for improving image generation quality within the image synthesis community. The authors have provided thorough theoretical analysis and experimental validation of PFSA in both their rebuttal and the appendix. We believe our concerns have been addressed and the work meets the standard for acceptance. Therefore, we assign a score of "5: Accept."

**Limitations:**

yes

**Quality:**

3

**Strengths And Weaknesses:**

Strengths:

(1) The paper proposes a plug-and-play, parameter-free self-attention mechanism (PFSA), which elegantly improves the global structural consistency of generated images at the cost of only a small increase in computational overhead. This effectively addresses a long-standing issue in Stable Diffusion when generating ultra-high-resolution images. If the method proves to be effective, it has the potential to become a general paradigm for high-resolution image generation tasks.

(2) The proposed RepLDM framework requires no additional training and features a simple, intuitive pipeline, making it easy to reproduce.

Weaknesses:

(1) As mentioned in Strengths (1), PFSA could be a promising approach if it truly works. However, due to its highly counter-intuitive nature, I find it difficult to judge whether it is genuinely effective. Why would a simple post-processing self-attention operation significantly improve so-called global structural consistency? Although the authors provide some supporting references (e.g., [44, 52]) and ablation studies (Section 5.1), the evidence remains unconvincing. A more detailed analysis of PFSA is needed—such as numerical evaluation or visualizations—to help readers better understand its mechanism. Moreover, the concept of global structural consistency should be more clearly defined.

(2) While the paper emphasizes high-efficiency in its title, the experimental section only compares inference time, which seems rather one-sided. A more comprehensive comparison including parameters, FLOPs, and NFE would provide a fuller picture of the method’s efficiency.

---

> ### Author Rebuttal · Authors · 2025-07-31
>
> We appreciate your kind words and recognition of the significance of our contributions. We did our best to address your questions as follows.
>
> > **W1: On the validation of PFSA’s effectiveness**
>
> We fully understand your concerns about the effectiveness of PFSA.
> To address them, we have conducted the following efforts:
>
> - In Section A of the supplementary material, we provide extensive analysis of the underlying mechanism of PFSA, which we elaborate on in our responses to Q3 and Q4.
> - We incorporate PFSA into ControlNet (see Section C of the appendix), demonstrating its applicability under different generation conditions.
> - We apply PFSA to various high-resolution training-free generation frameworks, including DemoFusion and HiDiffusion (Section D of the appendix), and observe consistent improvements, highlighting its generalizability.
> - We also integrate PFSA into Stable Diffusion 2.1, showing that PFSA is compatible with different versions of SD models.
> - Finally, we commit that, if the paper is accepted, we will release the RepLDM codebase promptly to facilitate reproducibility and further research within the community.
>
> > **W2: On the detailed explanation of global structural consistency**
>
> Great suggestion. In this paper, global structural consistency refers to the plausibility of the overall layout and the realism of object structures in an image. Specifically, a reasonable layout should follow logical spatial arrangements—for example, the sky should be above the ground. Realistic object structures should align with common sense, such as a cat having four legs instead of five. We will incorporate this explanation in the next version of the paper.
>
> > **W3: On the explanation of the effectiveness of Attention Guidance**
>
> To illustrate how Attention Guidance enhances the global structural consistency of latent representations, we conducted extensive experiments.
>
> 1. As shown in Fig. 12 of the supplementary material, we compute the deviation between the mean of each token and the overall mean across all tokens, and visualize this deviation. In columns (A) and (B) of Fig. 12, it can be observed that PFSA introduces pronounced stripe patterns in the token deviations, suggesting that it performs a clustering effect on tokens. Columns (C) and (D) present the heat maps of the deviation of tokens' mean, where it becomes evident that PFSA effectively aggregates semantically similar tokens, thereby enhancing the structural integrity and contours of the image.
> 2. To provide a more intuitive explanation, we conduct additional experiments to illustrate the effect of Attention Guidance. Specifically, given a clean RGB image $x$, we obtain its latent representation $z$ through VAE encoding.
> We then apply varying levels of noise to $z$:
> $z_t = \alpha_t \cdot \epsilon + (1 - \alpha_t) \cdot z$,
> where $t \in$ {$1,\ldots,9$}, $\alpha_t = 0.1\cdot t$, and $\epsilon \sim \mathcal{N}(0, I)$ is standard Gaussian noise.
> For $t=9,\ldots,1$, we compute $z_t^{PFSA}=PFSA(z_t)$ and $z_t^{AG} = \gamma_t \cdot z_t^{PFSA} + (1 - \gamma_t) \cdot z_t$, where $\gamma_9=1$ and gradually decays to $\gamma_1=0$ following the cosine annealing schedule.
> This process reflects the gradually diminishing effect of Attention Guidance (AG) during generation.
> We then decode $z_t$，$z_t^{PFSA}$, and $z_t^{AG}$ back to the RGB space, obtain $x_t$，$x_t^{PFSA}$, and $x_t^{AG}$, and compute their SSIM with respect to the original image $x$.
> The results are shown in the table below:
>
> | t | 9 | 8 | 7 | 6 | 5 | 4 | 3 | 2 | 1 |
> | --- | --- | --- | --- | --- | --- | --- | --- | --- | --- |
> | $x_t$ | 0.020 | 0.039 | 0.070 | 0.122 | 0.199 | 0.302 | 0.439 | 0.605 | **0.777** |
> | $x_t^{PFSA}$ | **0.038** | 0.066 | 0.083 | 0.106 | 0.110 | 0.120 | 0.153 | 0.205 | 0.279 |
> | $x_t^{AG}$ | **0.038** | **0.068** | **0.106** | **0.164** | **0.241** | **0.337** | **0.455** | **0.608** | **0.777** |
>
> It can be observed that directly applying PFSA improves SSIM in the early steps of denoising, but leads to a drop in SSIM in the middle and later steps. This explains why directly using PFSA could result in unstable denoising, as noted in Section 3.2.
> By incorporating the annealed strategy of Attention Guidance, we can leverage PFSA to promote structural recovery while avoiding numerical instability, thereby improving the visual quality of the generated images.
> Qualitative results demonstrate that Attention Guidance enables the more rapid removal of noise and the earlier unveiling of image structure. Due to policy constraints, we will provide the visual results in the next version of the paper. The results show that the smoothing effect of PFSA can suppress part of the noise, thereby enhancing structural similarity.
>
> A similar phenomenon can also be observed in a recent work, FreeU [Ref1], where image quality is improved by enhancing low-frequency components while suppressing high-frequency components. Interestingly, despite the suppression of high-frequency components, the generated images exhibit richer colors and details (as shown in their Figs. 10, 17, and 18). To sum up, these findings may suggest that reinforcing structural information during generation provides the model with a stronger prior for synthesizing the remaining fine details, thereby improving the overall visual quality of the images.
>
> [Ref1] FreeU: Free Lunch in Diffusion U-Net. CVPR2024.
>
> > **W4: On the numerical analysis of PFSA**
>
> To understand the effect of PFSA on the distribution of latent representations, we conducted a numerical analysis comparing latent features with and without the application of PFSA. Specifically, in Fig. 13 of Section A in the supplementary material, we compute the standard deviation and the Fourier amplitude spectrum of the latent representations with and without applying PFSA at different denoising timesteps.
> We observe that during the early and middle stages of denoising, PFSA increases the variance of the latent representations and amplifies the high-frequency components. In contrast, during the later stages, PFSA suppresses high-frequency components and reduces the standard deviation.
> This partially explains why attention guidance is more effective when applied during the early stages of denoising, but becomes less impactful in the later stages (as discussed in Fig. 9 of Section 5.1).
>
> > **W5: On the further visualizations for better understanding PFSA**
>
> According to your suggestion, we conducted an additional visualization experiment: We randomly selected a latent token as the query and visualized its attention weights as a heatmap over the generated RGB image to highlight the regions most related to the query token.
> The results show that even without the linear projection layers for queries and keys used in conventional self-attention, the query token can still identify the key tokens that are most semantically relevant to the query token. This explains why PFSA is able to cluster semantically related tokens and enhance global structural consistency, even in the absence of learnable linear layers.
>
> Unfortunately, we were informed via email that external links are not permitted in the rebuttal, and we are therefore temporarily unable to show these visualizations. However, we will incorporate the experiment in the next version of our paper.
>
> > **W6: On the efficiency comparison across different models**
>
> Great suggestion. To the best of our knowledge, existing training-free high-resolution generation frameworks [Ref1,Ref2,Ref3] evaluate efficiency solely based on inference time. This is primarily because these frameworks utilize the exact same pretrained model, ensuring identical parameters across comparisons. Nevertheless, we agree with your point that a more comprehensive set of metrics should be used to reflect efficiency. Following your suggestion, we have collected the NFE and FLOPs of different models when generating a single image at 2048 × 2048 and 4096 × 4096 resolutions. As all models employ training-free methods and have identical parameter counts, we do not report the number of parameters.
>
> Table 1: Inference cost of generating a 2048 × 2048 Image for different models.
>
> | Model | SDXL | MultiDiff. | ScaleCrafter | HiDiff. | UG | DemoFusion | AccDiffusion | RepLDM |
> | --- | --- | --- | --- | --- | --- | --- | --- | --- |
> | NFE | 50 | 50 | 50 | 50 | 80 | 100 | 100 | 60 |
> | TFLOPs | 3010 | 5420 | 2437 | 1857 | 3608 | 9015 | 8597 | 1140 |
> | Time (min) | 1.0 | 3.0 | 1.0 | 0.8 | 1.8 | 3.0 | 3.0 | 0.6 |
>
> Table 2: Inference cost of generating a 4096 × 4096 Image for different models.
>
> | Model | SDXL | MultiDiff. | ScaleCrafter | HiDiff. | UG | DemoFusion | AccDiffusion | RepLDM |
> | --- | --- | --- | --- | --- | --- | --- | --- | --- |
> | NFE | 50 | 50 | 50 | 50 | 80 | 200 | 200 | 65 |
> | TFLOPs | 12026 | 29566 | 9759 | 5211 | 12624 | 72167 | 74225 | 7140 |
> | Time (min) | 8.0 | 15.0 | 19.0 | 3.4 | 11.1 | 25.0 | 26.0 | 5.7 |
>
> Tables 1 and 2 show that RepLDM significantly reduces the NFE and FLOPs required for inference by decreasing the number of denoising steps at high resolutions, thereby substantially reducing the time needed to generate high-resolution images.
>
> [Ref1] AccDiffusion: An Accurate Method for Higher-Resolution Image Generation. ECCV 2024.\
> [Ref2] HiDiffusion: Unlocking Higher-Resolution Creativity and Efficiency in Pretrained Diffusion Models. ECCV2024.\
> [Ref3] DemoFusion: Democratising High-Resolution Image Generation With No . CVPR 2024.

---

> > ### Comment · Reviewer_urNj · 2025-08-01
> >
> > Thank you for your thoughtful response. We must admit that we overlooked the appendix during the review process. After your reminder, we carefully read the section on PFSA in the appendix and believe that you have adequately addressed our concerns. We will raise our rating to "accept."

---

> > > ### Author Response · Authors · 2025-08-05
> > >
> > > Dear Reviewer urNj,
> > >
> > > We are truly grateful for your detailed and insightful feedback on our paper. Your recognition and the updated score are deeply encouraging to us. Thank you again for your time and support.
> > >
> > > Warm regards,
> > >
> > > Authors of Paper 15032

---

### Official Review · Reviewer_ZoSx · 2025-06-27

**Clarity:** 3
**Significance:** 3
**Originality:** 3
**Rating:** 5
**Confidence:** 4

**Summary:**

The paper proposes a framework named RepLDM to reprogram pretrained latent diffusion models (LDMs) for generating high-resolution images without requiring additional training. The approach integrates a parameter-free self-attention mechanism to enhance the structural consistency of latent representations at the training resolution and progressive upsampling in pixel space to mitigate artifacts caused by latent space upsampling, improving both image quality and generation efficiency.

**Questions:**

1. PFSA's efficacy appears critically to be dependent on input latent quality.
(a) How does performance degrade under structured noise?
(b) Provide robustness analysis using metrics like SSIM drop ≥ 0.2 relative to baseline.

2. The complexity of PFSA's self-attention operation poses significant bottlenecks at high resolutions. Please give the computational load analysis for this issue.

3. For the progressive denoising with pixel space upsampling scheme,
(a)Does it rely on fixed-size convolutional kernels (e.g., 3×3/5×5) for pixel-space operations? Will it introduce the resolution issues when using different kernels?
(b) Did you explore integrating blind deblurring modules within the denoising steps?

**Ethical Concerns:**

["NO or VERY MINOR ethics concerns only"]

**Final Justification:**

After discussions with authors, I have decided to update the score.

**Limitations:**

Please see questions.

**Quality:**

3

**Strengths And Weaknesses:**

1.	The idea of reprogramming pretrained LDMs for high-resolution image generation is novel and resource-efficient.
2.	Upsampling in pixel space alleviates artifacts and allows for efficient denoising at higher resolutions, significantly reducing inference time.
3.	The paper conducts comprehensive quantitative and qualitative experiments to demonstrate the effectiveness and efficiency of RepLDM compared to state-of-the-art baselines.
Weaknesses:
1.	While RepLDM improves efficiency compared to other methods, generating ultra-high resolution images still requires more sub-stages in the progressive upsampling stage, potentially increasing inference time.
2.	The paper does not report error bars or statistical significance tests for the experimental results, making it difficult to assess the robustness of the claims.

---

> ### Author Rebuttal · Authors · 2025-07-31
>
> Thank you very much for your positive comments and efforts in reviewing our manuscript. We try our best to address your questions as follows.
>
> > **W1: On the efficiency of ultra-high-resolution image generation**
>
> For training-free image generation at ultra-high resolutions, the inference time is expected to increase further, a challenge commonly encountered by existing methods. However, our RepLDM still achieves a significant reduction in overall generation time compared to other frameworks, such as DemoFusion, despite employing a progressive upsampling sub-stage. This is because our RepLDM integrates an upsampling sub-stage within the entire denoising process over T timesteps, whereas in existing methods, the upsampling sub-stage is applied after each denoising process. We effectively reduce the number of denoising steps required at high resolutions, which are the primary source of computational overhead, as illustrated in Fig. 21 of the supplementary material.
>
> > **W2: On the error analysis of the quantitative metrics**
>
> Thank you for the constructive suggestion. Due to the high computational cost of high-resolution image generation, repeating experiments to compute standard deviations is expensive. As a result, most prior works do not perform error analysis [Ref1-Ref6]. To achieve a more rigorous evaluation, we conducted additional experiments to analyze the variability of the results. Specifically, for each experiment, we performed multiple generations (3 times) with different random seeds, computed the evaluation metrics for each generation, and reported the mean and standard deviation. Due to limitations in time and computational resources, we first performed this error analysis for RepLDM at a resolution of $2048 \times 2048$. We will extend this analysis to other resolutions and baseline models, and include the results in the next version of the paper.
> The results are shown in the table below:
>
> Table 1: Quantitative comparison results. The best results are marked in **bold**, and the second best results are marked by $\underline{\text{underline}}$.
> | Model | $\text{FID}$ | $\text{IS}$ | $\text{FID}_c$ | $\text{IS}_c$ | $\text{CLIP}$ |
> | --- | --- | --- | --- | --- | --- |
> | SDXL | $99.94$ | $14.22$ | $80.00$ | $16.91$ | $25.03$ |
> | MultiDiff. | $98.83$ | $14.54$ | $67.87$ | $17.14$ | $24.59$ |
> | ScaleCrafter | $98.18$ | $14.19$ | $89.71$ | $13.32$ | **25.37** |
> | UG | $82.22$ | $17.61$ | $65.81$ | $14.64$ | $25.52$ |
> | HiDiff. | $81.01$ | $16.79$ | $64.14$ | $14.15$ | $24.94$ |
> | DemoFusion | $72.29$ | $\underline{21.62}$ | $53.49$ | $\underline{19.11}$ | $25.21$ |
> | AccDiff. | $\underline{71.63}$ | $21.00$ | $\underline{52.65}$ | $17.00$ | $25.13$ |
> | RepLDM | **66.08**$\pm$0.02 | **22.13**$\pm$0.74 | **47.31**$\pm$0.11 | **20.38**$\pm$2.03 | $\underline{25.30}\pm 0.12$ |
>
> The results show that FID, FID_c, and CLIP exhibit relatively small variance, while IS shows comparatively larger fluctuations. This may be attributed to the fact that IS is not well-suited for evaluation on datasets other than ImageNet [Ref7]. In our future work, we will adopt more reliable metrics, such as KID, to replace IS for more accurate evaluation [Ref8].
>
> [Ref1] DemoFusion: Democratising High-Resolution Image Generation With No. CVPR2024.\
> [Ref2] AccDiffusion: An Accurate Method for Higher-Resolution Image Generation. ECCV2024.\
> [Ref3] FouriScale: A Frequency Perspective on Training-Free High-Resolution Image Synthesis. ECCV2024.\
> [Ref4] ScaleCrafter: Tuning-free Higher-Resolution Visual Generation with Diffusion Models. ICLR2024.\
> [Ref5] Training-free Diffusion Model Adaptation for Variable-Sized Text-to-Image Synthesis. NeurIPS2023.\
> [Ref6] HiDiffusion: Unlocking Higher-Resolution Creativity and Efficiency in Pretrained Diffusion Models. ECCV2024.\
> [Ref7] A Note on the Inception Score. ICML2018.\
> [Ref8] Demystifying MMD GANs. ICLR2018.
>
> > **Q1: On the efficacy of PFSA**
>
> Thank you for your suggestion. We would like to note that PFSA itself does not contain any parameters, and thus its efficacy is not affected by the input. Notably, PFSA influences the efficacy of Attention Guidance through the parameter $\gamma$, thereby altering the generation results. Therefore, $\gamma$ is a key parameter that significantly impacts generation quality.
> Furthermore, for evaluation metrics, while the SSIM metric is frequently employed in image super-resolution, our study primarily focuses on text-to-image generation, where ground truth images are unavailable.
> In this paper, we utilize commonly adopted metrics in generation tasks—FID, IS, and CLIP—to comprehensively assess image quality, consistent with previous methods. In addition, considering that our work targets high-resolution image generation, we follow prior works [Ref1,Ref2] and compute FID and IS on cropped patches of high-resolution images, denoted as $\text{FID}_c$ and $\text{IS}_c$.
> The results in Table 1 demonstrate the robustness of our RepLDM.
> In Section 5.1, we conduct both qualitative and quantitative ablation studies on $\gamma$, and provide a more comprehensive analysis in Section F.1 of the supplementary material.
> Our ablation results suggest that, within a certain range, larger values of $\gamma$ tend to introduce more fine-grained details and lead to better visual quality in the generated results.
>
> [Ref1] DemoFusion: Democratising High-Resolution Image Generation With No $$$. CVPR2024.\
> [Ref2] AccDiffusion: An Accurate Method for Higher-Resolution Image Generation. ECCV2024.
>
> > **Q2: On the computational complexity analysis of PFSA**
>
> Great suggestion. We would like to note that PFSA is only applied during the first stage of generation, and therefore does not introduce any computational or memory bottlenecks.
> Assume we have a high-resolution image $x$ with a resolution of $H \times W \times C$.
> we encode the image $x$ into latent space and obtain latent representation $z \in \mathbb{R}^{h \times w \times c}$.
> Before feeding $z$ into PFSA, we reshape it to a $(hw) \times c$ matrice.
> The computation of PFSA follows a formulation similar to that of self-attention:\
> $\text{PFSA}(z)=\text{Softmax}(\frac{zz^T}{\sqrt{c}})z$\
> Thus, the computational complexity of PFSA is $O((hw)^2c)$.
> Taking SDXL as an example, the training resolution is $H=1024$, $W=1024$.
> After VAE encoding, $c=4$, $h=H/8=128$, $w=W/8=128$.
> For each denoising step, the FLOPs of PFSA is approximately $2 \times (h \times w)^2 \times c$, which is around 2.15 GFLOPs—negligible compared to the FLOPs of the denoising network (several TFLOPs per step).
>
> > **Q3: On the pixel space upsampling scheme**
>
> (a) The proposed RepLDM avoids artifacts caused by upsampling in the latent space by decoding latent representations into the pixel space for upsampling.
> In practice, we use bicubic interpolation to perform this upsampling, which does not rely on a convolutional kernel of fixed size. Alternatively, a super-resolution network can be employed to accomplish this upsampling step, potentially leading to better generation quality.\
> (b) We believe your proposed approach is insightful and highly feasible. We have previously considered a similar idea—training a neural network to directly perform upsampling in the latent space—which may align with your suggestion of using a blind model to mitigate the blurring artifacts caused by latent-space interpolation. We find your suggestion very constructive. To train such a blind model, a sufficiently large and high-quality high-resolution dataset would be required. Due to the limited time during the rebuttal phase, we will leave this as a promising direction for future exploration.

---

> ### Author Response · Authors · 2025-08-05
>
> Dear Reviewer ZoSx,
>
> Thank you once again for your comprehensive and thoughtful feedback on our submission. As the discussion period nears its end, we are eager to know if our additional results and clarifications have adequately addressed your questions. We would sincerely appreciate any further perspectives or discussions you might have at this stage. Thank you for your time and engagement!
>
> Best regards,
>
> Authors of Paper 15032

---

> > ### Comment · Reviewer_ZoSx · 2025-08-06
> >
> > Thank you for your efforts. The detailed response has partially addressed my concerns, and I have updated my previous rating.

---

> > > ### Author Response · Authors · 2025-08-06
> > >
> > > Dear Reviewer ZoSx,
> > >
> > > Thank you once again for your thoughtful review and the encouraging feedback you provided. We sincerely appreciate your time and insights.
> > >
> > > We noticed that the decision status for our paper has not yet been updated. We wanted to kindly check in to confirm whether there might have been a delay in reflecting the decision or if any further action is needed on our part.
> > >
> > > Thank you for your attention to this matter——we truly value your guidance.
> > >
> > > Warm regards,
> > >
> > > The Authors of Paper 15032

---

### Official Review · Reviewer_6jTA · 2025-06-27

**Clarity:** 3
**Significance:** 3
**Originality:** 4
**Rating:** 5
**Confidence:** 4

**Summary:**

The authors propose a method for sampling high-resolution images using pretrained diffusion models. Their method consists of progressively making the image larger as they perform the denoising process. Additionally, to better capture the image details they make use of a parameter free self attention operator that shows promise in improving sample quality.

**Questions:**

My biggest question is trying to understand what attention guidance is doing. Although the authors provided some discussion on the appendix, it is still not clear to me. For ease of this discussion lets consider a black and white image, therefore we only have one channel. Although I understand that this operation is done in the latent space the concept should be the same.

When using attention guidance $$AG(v) = \sigma(vv^T)v$$ the values of v become close to each other and not the other way around. Therefore when we perform the operation $$v  = \gamma AG(v) + (1-\gamma) v = v + \gamma (AG(v) - v)) $$ the pixels with higher values (more black) become smaller and white values become smaller. Therefore the whole image is converging to a more smoothened gray image.

Analogously this operation in the latent space pulls the "latent pixels" closer to each other, which should reduce the details. Therefore that would suggest that there is something particular about the latent space that AG is using to make this work. Could you elaborate on this?

Additionally, it would be nice to provide a discussion on why do the numerical results are so similar while the qualitative samples look so different

My final question is regarding the experiments. It seems that you have limited compute and perhaps this could be a cause of issue, but FID and other metrics like IS have very high variance when using a small amount of samples like $1000$. Additionally it seems like the reference statistics are from the SAM dataset, which are not related to the aesthetics generation that SDXL and these models are trained on, therefore the results are a little hard to interpret. Although the qualitative examples do show lots of promise, it would be good to have this be a more robust metric

**Ethical Concerns:**

["NO or VERY MINOR ethics concerns only"]

**Final Justification:**

Sampling images from diffusion models in high dimensions has remained a very challenging problem. This paper presents a way to do it that qualitatively is producing remarkable results. Although some of the quantitative metrics don't seem to be done in the best way, I believe that it is a great first step towards sampling images in high dimensions.

**Limitations:**

yes

**Paper Formatting Concerns:**

-

**Quality:**

3

**Strengths And Weaknesses:**

Strengths
- The paper provides a clear explanation of the method
- The paper provides lots of experiments and comparisons

Weaknesses

- Could the authors define HR and TR. I believe I know there meaning but it would be good for the paper
- The method requires decoding and encoding multiple times, which is not computationally ideal

---

> ### Author Rebuttal · Authors · 2025-07-31
>
> Thank you very much for your insightful comments and suggestions. We try our best to address your questions as follows.
>
> > **W1: On the definitions of training resolution and high resolution**
>
> Great suggestion. To improve clarity, we will refine the annotation of the abbreviations HR and TR, and include the following explanation as a footnote in the next revision: In this paper, training resolution refers to the resolution used during model training; high resolution refers to a resolution that significantly exceeds the training resolution, to the extent that the model cannot directly generate satisfactory results at that resolution.
>
> > **W2: On the efficiency analysis of intermediate encoding-decoding operations**
>
> To evaluate the computational cost introduced by multiple encoding and decoding operations, we analyze the time consumption of each component in DemoFusion and RepLDM when generating images at the resolution of $4096 \times 4096$.
> The results are shown in the table below:
>
> Table 1: The time consumption of DemoFusion when generating 4096×4096 resolution images.
> | Metric | Denoise 1024 | Denoise 2048 | Denoise 3072 | Denoise 4096 | Decode 4096 | Total |
> | --- | --- | --- | --- | --- | --- | --- |
> | num of steps | $50$ | $50$ | $50$ | 50 | / | $200$ |
> | Time (s) | $12$ | $185$ | $480$ | $901$ | $106$ | $1684$ |
>
> Table 2: The time consumption of RepLDM when generating 4096×4096 resolution images. The intermediate encoding/decoding operations are highlighted in $\underline{\text{underline}}$.
> |  | Denoise 1024 | Decode 1024 | Encode 3304 | Denoise 3304 | Decode 3304 | Encode 4096 | Denoise 4096 | Decode 4096 | Total |
> | --- | --- | --- | --- | --- | --- | --- | --- | --- | --- |
> | num of steps | $50$ | / | / | $5$ | / | / | $10$ | / | $65$ |
> | Time (s) | $12$ | $\underline{0}$ | $\underline{12}$ | $20$ | $\underline{64}$ | $\underline{11}$ | $118$ | $106$ | $343$ |
>
> As shown in Table 1, denoising at high resolutions is a time-consuming process.
> DemoFusion requires substantial generation time because it performs the full denoising process at high resolutions.
> Table 2 shows that RepLDM significantly accelerates generation by substantially reducing the number of denoising steps at high resolutions.
> This is because RepLDM performs pixel-space upsampling through multiple rounds of encoding and decoding, producing high-quality low-resolution images that serve as better initialization. As a result, RepLDM can significantly reduce the number of sampling steps required for high-resolution generation, thereby accelerating the process. Moreover, Table 2 shows that the additional overhead from multiple intermediate encoding and decoding operations is relatively minor compared to the total generation cost.
>
> > **Q1: On the analysis of the effectiveness of Attention Guidance**
>
> Very insightful comment. To provide an intuitive explanation, we conduct additional experiments to illustrate the effect of Attention Guidance (AG). Specifically, given a clean RGB image $x$, we obtain its latent representation $z$ through VAE encoding.
> We then apply varying levels of noise to $z$:
> $z_t = \alpha_t \cdot \epsilon + (1 - \alpha_t) \cdot z$,
> where $t \in$ {$1,\ldots,9$}, $\alpha_t = 0.1\cdot t$, and $\epsilon \sim \mathcal{N}(0, I)$ is standard Gaussian noise.
> For $t=9,\ldots,1$, we compute $z_t^{PFSA}=PFSA(z_t)$ and $z_t^{AG} = \gamma_t \cdot z_t^{PFSA} + (1 - \gamma_t) \cdot z_t$, where $\gamma_9=1$ and gradually decays to $\gamma_1=0$ following the cosine annealing schedule.
> This operation reflects the progressively diminishing influence of Attention Guidance during the generation process.
> We then decode $z_t$，$z_t^{PFSA}$, and $z_t^{AG}$ back to the RGB space, obtain $x_t$, $x_t^{PFSA}$, and $x_t^{AG}$, and compute their SSIM with respect to the original image $x$.
> The results are shown in the table below:
>
> Table 3: The SSIM values of $x_t$, $x_t^{PFSA}$, and $x_t^{AG}$ with respect to the original image $x$. The best results are highlighted in **bold**.
> | t | 9 | 8 | 7 | 6 | 5 | 4 | 3 | 2 | 1 |
> | --- | --- | --- | --- | --- | --- | --- | --- | --- | --- |
> | $x_t$ | 0.020 | 0.039 | 0.070 | 0.122 | 0.199 | 0.302 | 0.439 | 0.605 | **0.777** |
> | $x_t^{PFSA}$ | **0.038** | 0.066 | 0.083 | 0.106 | 0.110 | 0.120 | 0.153 | 0.205 | 0.279 |
> | $x_t^{AG}$ | **0.038** | **0.068** | **0.106** | **0.164** | **0.241** | **0.337** | **0.455** | **0.608** | **0.777** |
>
> It can be observed that directly applying PFSA improves SSIM in the early steps of denoising, but leads to a drop in SSIM in the middle and later steps.
> This explains why directly using PFSA could result in unstable denoising, as noted in Section 3.2.
> By incorporating the annealed strategy of Attention Guidance, we can leverage PFSA to promote structural recovery while avoiding numerical instability, thereby improving the visual quality of the generated images.
> Qualitative results demonstrate that Attention Guidance enables the more rapid removal of noise and the earlier unveiling of image structure. Due to policy constraints, we will provide the visual results in the next version of the paper. The results show that the smoothing effect of PFSA can suppress part of the noise, thereby enhancing structural similarity.
>
> A similar phenomenon can also be observed in a recent work, FreeU [Ref1], where image quality is improved by enhancing low-frequency components while suppressing high-frequency components. Interestingly, despite the suppression of high-frequency components, the generated images exhibit richer colors and details (as shown in their Figs. 10, 17, and 18). To sum up, these findings may suggest that reinforcing structural information during generation provides the model with a stronger prior for synthesizing the remaining fine details, thereby improving the overall visual quality of the images.
>
> [Ref1] FreeU: Free Lunch in Diffusion U-Net. CVPR2024.
>
> > **Q2: On the similar quantitative ablation results of Attention Guidance**
>
> Thank you for your suggestion. We would like to point out that FID and IS quantify the statistical differences between two distributions [Ref1,Ref2,Ref3]. Since Attention Guidance mainly enhances visual quality by modifying the mid- and high-frequency components while preserving the low-frequency structure of the image, it has limited impact on the overall distributional statistics. Consequently, the resulting quantitative metrics remain similar.
>
> [Ref1] A Note on the Inception Score. ICML2018.\
> [Ref2] Improved Techniques for Training GANs. NeurIPS2016.\
> [Ref3] Gans trained by a two time-scale update rule converge to a local nash equilibrium. NeurIPS2017.
>
> > **Q3: On the evaluation metrics and datasets**
>
> During our experiments, Laion-5B was removed from Huggingface due to potential content risks. We note that the SAM dataset is of high resolution (average: $3300 \times 4950$). Although the released version is a downsampled version with the shorter side resized to 1500, the resolution still remains significantly higher than that of many existing vision datasets (e.g., COCO) [Ref1]. Therefore, we chose the SAM dataset as a substitute. For the evaluation metrics and the number of generated samples used to evaluate performance, we follow the settings used in several related works, where 1k prompts are used to generate $4096 \times 4096$ images for evaluation [Ref2]. This choice is made due to the extremely high inference cost associated with generating high-resolution images.
>
> We highly appreciate your constructive suggestions regarding the experiments. For a more rigorous comparison, we conducted additional experiments:
>
> - We recently discovered the availability of high-resolution Laion images on Huggingface and downloaded 20K images to serve as the benchmark.
> - To enable more stable evaluation, we doubled the number of prompts used to generate $4096 \times 4096$ resolution images. Due to the substantial time required to generate 4096 × 4096 resolution images and current hardware constraints, we will provide the relevant experimental results in the next version of the paper.
> - For RepLDM, we performed multiple generations (3 times) at 2048×2048 resolution using different random seeds and computed the mean and standard deviation. Due to time and hardware constraints, we will conduct the same variance analysis for other models as well.
> - Regarding evaluation metrics, since Inception Score (IS) may lead to high variances beyond ImageNet, we follow prior work and adopt Kernel Inception Distance (KID) for more accurate evaluation [Ref6,Ref7].
>
> Due to time and hardware constraints, we have temporarily compared our RepLDM with the top three baselines from Table 1 of the paper: DemoFusion, AccDiffusion, and HiDiffusion.
> We will continue to extend the experiments and include the results and analyses in the next version of the paper.
> The results are shown in the table below:
>
> Table 4: **Quantitative comparison results at $2048 \times 2048$ resolution**. The best results are marked in **bold**.
> | Model | $\text{FID}$ | $\text{KID}$ | $\text{FID}_c$ | $\text{KID}_c$ | $\text{CLIP}$ |
> | --- | --- | --- | --- | --- | --- |
> | HiDiffusion | 48.73 | 0.0083 | 36.73 | 0.0111 | $23.19$ |
> | AccDiffusion | $34.91$ | $0.0050$ | $22.60$ | $0.0085$ | $24.93$ |
> | DemoFusion | $34.53$ | $0.0046$ | $21.36$ | $0.0067$ | $25.46$ |
> | RepLDM | **34.08**$\pm$0.25 | **0.0042**$\pm$0.0000 | **20.30**$\pm$0.30 | **0.0049**$\pm$0.0001 | **25.78**$\pm$0.03 |
>
> Table 4 shows that RepLDM still achieves state-of-the-art performance on the Laion dataset, further highlighting its strong generalization capability.
>
> [Ref1] Segment Anything. ICCV2023.\
> [Ref2] DemoFusion: Democratising High-Resolution Image Generation With No . CVPR2024.\
> [Ref3] A Note on the Inception Score. ICML2018.\
> [Ref4] ScaleCrafter: Tuning-free Higher-Resolution Visual Generation with Diffusion Models. ICLR2024.\
> [Ref5] Demystifying MMD GANs. ICLR2018.

---

> > ### Comment · Reviewer_6jTA · 2025-08-02
> >
> > I thank the authors for their answers. In view of effort and importance of the work (sampling diffusion in high dimension remains challenging) I have increased my score.

---

> > > ### Author Response · Authors · 2025-08-05
> > >
> > > Dear Reviewer 6jTA,
> > >
> > > Thank you very much for taking the time to review our work and for your thoughtful and constructive feedback throughout the process. We sincerely appreciate your recognition of our contributions and your updated evaluation — it means a lot to us.
> > >
> > > Best regards,
> > >
> > > Authors of Paper 15032

---

### Official Review · Reviewer_46id · 2025-07-01

**Clarity:** 3
**Significance:** 3
**Originality:** 3
**Rating:** 5
**Confidence:** 4

**Summary:**

This paper introduces RepLDM, a novel training-free framework for adapting pretrained latent diffusion models (LDMs) for high-resolution image generation. The method tackles the common problem of structural distortion when generating images at resolutions beyond the model's training data. RepLDM works in two stages: (1) an Attention Guidance stage that uses a novel parameter-free self-attention (PFSA) mechanism to generate a structurally-consistent latent at the base training resolution, and (2) a Progressive Upsampling stage that iteratively increases the image resolution in pixel space to avoid artifacts common to latent-space upsampling. By starting the upsampling process with a high-quality latent, the method requires significantly fewer denoising steps at high resolutions, leading to major gains in both image quality and inference speed.

**Questions:**

- Your formulation for PFSA effectively sets the Query, Key, and Value representations equal to the same feature representation $f(z)$. This is a fascinating simplification. Have you considered or experimented with asymmetric variants within the parameter-free constraint? For example, could one use the raw feature $f(z)$ for the Query, but a structurally-simplified version (e.g., from a blurred or pooled feature map) for the Key and Value? My hypothesis is that this might allow the model to more explicitly attend *from* fine details *to* coarse structures.

- I recently came across a work titled "Swift parameter-free attention network for efficient super-resolution" which also explores parameter-free attention mechanisms before. Could you clarify how your Parameter-Free Self-Attention (PFSA) mechanism relates to or differs from the approach in that work? This would help to better position your contribution within the existing literature.

- You note that directly applying PFSA can lead to unstable denoising, which motivates the linear interpolation with the original latent $z$ in \textbf{Eq. 2}. Could you offer more insight into this instability? For instance, does $PFSA(z)$ produce a latent with a significantly different statistical distribution (e.g., higher variance) compared to $z$? Is the primary role of the $(1-\gamma)z$ term to act as a statistical anchor to keep the guided latent on-manifold?

- The use of a cosine annealing schedule for $\gamma_{t}$ is a well-motivated choice, aligning guidance strength with the diffusion model's tendency to generate structure first. How sensitive is the final image quality to the specific functional form of this schedule? Did you experiment with simpler schedules, like linear or step-wise decay, and did they yield comparable results?

- The experiments are thoroughly performed using SDXL as the base model. How well do the RepLDM framework and its associated hyperparameters ($\gamma$, $\eta_{1}$, $\eta_{2}$) generalize to other popular pretrained LDMs, such as the Stable Diffusion v1.5/v2.1 series or various community fine-tuned models?

**Ethical Concerns:**

["NO or VERY MINOR ethics concerns only"]

**Final Justification:**

Clarifications on the “parameter-free” terminology, additional experiments exploring asymmetric PFSA, and the new annealing-schedule and stability analyses directly address my main questions. I appreciate the side-by-side comparisons (e.g., Table A-sym PFSA) and the evidence of hyper-parameter robustness across SD v2.1, ControlNet, DemoFusion, and HiDiffusion. All remaining issues are minor (terminology and small wording edits). I therefore maintain my overall positive assessment and still recommend acceptance. I look forward to seeing the updated manuscript incorporating the promised revisions and additional results.

**Limitations:**

Yes. The authors provide a clear and well-written "Limitations and Future Work" section (Section 6). They responsibly acknowledge the method's inability to fix inherent flaws in the base model (like text generation) and discuss the computational scaling challenges when targeting ultra-high resolutions. This transparent reporting strengthens the paper.

**Paper Formatting Concerns:**

None.

**Quality:**

3

**Strengths And Weaknesses:**

Strengths:
- Novelty and Elegance of Core Theory (PFSA): The paper's central contribution, the \textbf{Parameter-Free Self-Attention (PFSA)} mechanism, is exceptionally elegant. The formulation in \textbf{Eq. 1} ($PFSA(z) = f^{-1}(\text{Softmax}(\frac{f(z)f(z)^{T}}{\lambda})f(z))$) cleverly strips the standard self-attention mechanism of its learnable projection matrices ($W_{Q}, W_{K}, W_{V}$). This design is based on the strong and insightful premise that the global spatial modeling capability of attention is inherent to its structure, not its learned parameters. This simplification provides a powerful, first-principles approach to enhancing structural consistency in a training-free manner.

- The decomposition of the problem into an "Attention Guidance" stage and a "Progressive Upsampling" stage is logical and effective. This design smartly separates the task of enforcing global structure (at the native resolution) from the task of adding details at higher resolutions. The framework is clean and addresses two distinct failure modes of naive high-resolution generation.

- The authors' identification of latent-space upsampling as a key source of artifacts is a critical insight. Their pilot study (Fig. 4) provides clear empirical evidence for this hypothesis. The decision to perform upsampling in the pixel domain, as formalized in Equation 4, $\hat{z}_{0} = \mathcal{E} \circ \mathcal{U} \circ \mathcal{D}(z_{0})$, is a simple but highly effective solution that directly mitigates this issue, demonstrating a deep understanding of the VAE's properties and limitations.

- The method delivers on its promises, significantly outperforming strong recent baselines like DemoFusion and AccDiffusion across multiple metrics (FID, IS, CLIP Score) and resolutions. The ~5x speedup in inference time is a massive practical advantage that makes high-resolution generation far more accessible. The user study further corroborates the superior quality of the generated images.

Weaknesses
- Nuance in "Parameter-Free" Claim- While the core PFSA block is indeed parameter-free, the overall attention guidance mechanism introduces several key hyperparameters that require careful tuning. The linear interpolation in Equation 2 depends on the guidance scale $\gamma$, and the adaptive schedule in Equation 3 introduces a decay factor $\beta$ and a delay rate $\eta_{1}$. While these are not learned weights, they represent a set of sensitive parameters that are crucial to the method's success, making the "parameter-free" description of the overall guidance slightly imprecise.

- As the authors rightly point out in Section 6, the method is a "reprogramming" framework, not a fine-tuning one. As such, it cannot fix fundamental flaws of the base LDM, such as the difficulty with generating coherent text. This is an inherent limitation of the approach.

---

> ### Author Rebuttal · Authors · 2025-07-31
>
> We are grateful for your positive summary and constructive suggestions. We have tried our best to address your concerns as follows.
>
>  > **W1:  On the definition of “parameter-free”**
>
> Good catch! In our paper, the term “parameter-free” in PFSA was intended to emphasize the absence of learnable parameters, in contrast to vanilla self-attention with learnable parameters. We appreciate your suggestion and will adopt a more accurate terminology in the next version by replacing *Parameter-Free Self-Attention (PFSA)* with *Training-Free Self-Attention (TFSA)*.
>
> >**W2: On the fundamental flaws of the base LDM**
>
> We would like to clarify this point from three perspectives.
> - We would like to note that training-free frameworks are inherently limited by the capabilities of their underlying base models, which is a widely observed issue. To the best of our knowledge, existing training-free generation frameworks have not effectively addressed this limitation. Some efforts to address the limitations of base models, such as improving fine-grained text generation, rely on fine-tuning the base model with specific datasets [Ref1,Ref2,Ref3].
> - Although fine-tuning the base model can alleviate some of its inherent limitations, it may come at the cost of degrading its generative quality. In comparison, the reprogramming framework better retains the base model’s original generation capacity.
> - Our proposed RepLDM is a general framework for high-resolution generation. In Section C of the supplementary material, we demonstrate that RepLDM is also compatible with different generation conditions. This suggests that RepLDM can be applied to alternative, improved base models to circumvent certain limitations of the original base model.
>
> [Ref1] AnyText: Multilingual Visual Text Generation And Editing. ICLR2024.\
> [Ref2] TextDiffuser-2: Unleashing the Power of Language Models for Text Rendering. ECCV2024.\
> [Ref3] TextDiffuser: Diffusion Models as Text Painters. NeurIPS 2023.
>
> > **Q1: On the different PFSA calculation paradigm**
>
> For PFSA, our objective is to remove the learnable parameters from the Self-Attention mechanism, while maintaining its computational paradigm as unchanged as possible. Therefore, we opted to use symmetric $Q$ and $K$ matrices for computing PFSA. We appreciate your suggestion of using asymmetric $Q$ and $K$ matrices for PFSA computation, which we find to be a very interesting idea. To verify its effectiveness, we conducted additional experiments accordingly. Specifically, we apply a 2 $\times$ 2 pooling operation to downsample the $K$ and $V$ matrices. This ensures that the output of $\text{Softmax}(QK^T/\sqrt{d})V$ remains of shape $(hw) \times c$. We then use this new output tensor for guidance, while keeping all other hyperparameters unchanged. The results at 2048 $\times$ 2048 resolution are shown in the table below (the best results are marked in **bold**):
>
> | Method | $\text{FID}$ | $\text{IS}$ | $\text{FID}_c$ | $\text{IS}_c$ | $\text{CLIP}$ |
> | --- | --- | --- | --- | --- | --- |
> | w/o PFSA | 66.8 | 21.6 | 47.5 | 17.4 | **25.3** |
> | Asymmetric PFSA | 67.4 | **22.6** | 47.9 | **20.4** | **25.3** |
> | PFSA | **66.0** | 21.0 | **47.4** | 17.5 | 25.1 |
>
> Quantitatively, all metrics, except for FID, show improvement. Qualitatively, compared to PFSA, asymmetric PFSA results in fewer details but demonstrates more advantages in structural adjustments. Due to conference policies, we are unable to provide access to these qualitative results at this time; however, we will include these analyses and experiments in the next version of the paper.
>
> By the way, we have considered alternative attention paradigms for guidance, such as parameter-free linear attention mechanisms. However, our experiments showed that using a linear attention mechanism for guidance directly leads to meaningless noise or entirely black images.
>
> > **Q2: On the difference between our PFSA and PFAM proposed by Wan et al.**
>
> We appreciate your recommendation and have read the paper. In the research conducted by Wan et al., they proposed a parameter-free attention mechanism (PFAM) to replace self-attention for more efficient image super-resolution.
> PFAM differs from the conventional Self-Attention in that it does not compute the similarity between tokens. Instead, the attention map in PFAM is obtained by applying the proposed symmetric activation function to the output of the convolutional layer. The output of PFAM is the result of an element-wise multiplication between the attention map and the features.
>
> Overall, our PFSA differs from their attention mechanism (PFAM) in the following aspects:
>
> - Task objective: PFSA is designed for generative tasks, aiming to improve the quality of high-resolution outputs. In contrast, PFAM focuses on super-resolution, where maintaining consistency between high- and low-resolution images is the primary goal.
> - Modeling approach: PFSA retains the computational paradigm of self-attention and can be considered a specialized form of self-attention. In contrast, PFAM does not compute token similarity; instead, it focuses on the magnitude of tensor elements. Conceptually, its attention mechanism differs quite significantly from that of Self-Attention.
> - Training strategy: PFSA operates without any training; although PFAM is parameter-free, it still requires joint training with convolutional layers to be effective.
>
> > **Q3: On the stability analysis of PFSA and the role of $(1 - \gamma)z$**
>
> Very insightful comment. First, we would like to draw your attention to Section A of the supplementary material, where we provided a stability analysis of the PFSA.
> Specifically: (**i**) We compute the deviation of token means and observe that PFSA exhibits a clustering effect, bringing semantically related tokens closer together (Fig. 12).
> (**ii**) We compute the standard deviation of latent representations and their Fourier amplitude spectra, and find that PFSA is capable of modulating both the amplitudes across different frequency bands and the standard deviation of latent representations (Fig. 13).
>
> These analyses suggest that PFSA may operate by modulating the distribution of latent representations. To make this effect more gradual and controlled, we introduce a guidance mechanism to adjust the representations progressively.
>
> Your observation is highly insightful — the use of a linear combination with $(1 - \gamma)z$ serves to anchor the guided latent representations, helping to ensure that the latent representations remain on the data manifold. We will incorporate the explanation from the manifold perspective you mentioned in Section 3.2 of the next version of the paper.
>
> > **Q4: On the sensibility analysis of annealing strategy**
>
> Great suggestion. Previous studies have shown that diffusion models first generate low-frequency structures and then generate fine details [Ref4,Ref5,Ref6]. Based on this observation, we introduce the annealing strategy to gradually decrease the structural guidance effect of PFSA. Considering that cosine annealing has demonstrated robustness in various applications (e.g., learning rate decay), we employ cosine annealing in RepLDM.
> We appreciate your constructive suggestion. Following your recommendation, we performed additional experiments using linear and exponential annealing strategies at 2048 $\times$ 2048 resolution to investigate the sensitivity of Attention Guidance to the choice of annealing strategy.
> The results are shown in the table below:
>
> | Method | $\text{FID}$ | $\text{IS}$ | $\text{FID}_c$ | $\text{IS}_c$ | $\text{CLIP}$ |
> | --- | --- | --- | --- | --- | --- |
> | Linear | 66.2 | 21.5 | 47.2 | 20.3 | 25.4 |
> | exponential | 66.8 | 21.8 | 47.0 | 16.3 | 25.3 |
> | cosine | 66.0 | 21.0 | 47.4 | 17.5 | 25.1 |
>
> Quantitatively, comparable performance is observed across different annealing strategies. Qualitatively, the generated results also exhibited comparable improvements in structure and details. This suggests that Attention Guidance is not sensitive to a specific annealing strategy. Due to conference policy restrictions, we are unable to provide access to the qualitative analysis at this time, but we will include these analyses in the next version of the paper.
>
> [Ref4] FreeEnhance: Tuning-Free Image Enhancement via Content-Consistent Noising-and-Denoising Process. ACM MM2024.\
> [Ref5] Relay Diffusion: Unifying diffusion process across resolutions for image synthesis. ICLR2024.\
> [Ref6] FreeDoM: Training-Free Energy-Guided Conditional Diffusion Model.
>
> > **Q5: On the generalization of hyperparameters to other pretrained LDMs**
>
> Great questions! We would like to draw your attention to Section E of Supplementary Material. To verify the generalization of the hyperparameters, we conducted experiments using the guidance hyperparameter settings of SDXL to Stable Diffusion 2.1. The results demonstrate that the hyperparameters used for SDXL are also applicable to SD2.1.
> Furthermore, we tested the same settings on ControlNet and other training-free generative frameworks (HiDiffusion, DemoFusion), with all results confirming the robustness of this hyperparameter set.

---

> ### Comment · Reviewer_46id · 2025-08-02
>
> Thank you for the thorough and thoughtful rebuttal. Your clarifications on the “parameter-free” terminology, additional experiments exploring asymmetric PFSA, and the new annealing-schedule and stability analyses directly address my main questions. I appreciate the side-by-side comparisons (e.g., Table A-sym PFSA) and the evidence of hyper-parameter robustness across SD v2.1, ControlNet, DemoFusion, and HiDiffusion. All remaining issues are minor (terminology and small wording edits). I therefore maintain my overall positive assessment and still recommend acceptance. I look forward to seeing the updated manuscript incorporating the promised revisions and additional results. Thus, I will raise to accept.

---

> > ### Author Response · Authors · 2025-08-05
> >
> > Dear Reviewer 46id,
> >
> > Thank you once again for your thoughtful review and the encouraging feedback you provided. We sincerely appreciate your time and insights.
> >
> > We noticed that the decision status for our paper has not yet been updated. We wanted to kindly check in to confirm whether there might have been a delay in reflecting the decision or if any further action is needed on our part.
> >
> > Thank you for your attention to this matter——we truly value your guidance.
> >
> > Warm regards,
> >
> > The Authors of Paper 15032

---

> > > ### Comment · Reviewer_46id · 2025-08-05
> > >
> > > I've just updated it; please check that.

---

### Note · Authors · 2025-08-15

We sincerely thank all reviewers for their thorough reviews and constructive feedback. We are encouraged that the reviewers recognize RepLDM's innovations in efficient high-resolution image generation, practical value for resource-limited scenarios. According to the reviewers’ suggestions, we incorporate the following key improvements and clarifications in the revised paper:

> **Clarifications of Definitions and Annotations**

1. We replace the term Parameter-Free Self-Attention (PFSA) with the more accurate Training-Free Self-Attention (TFSA). Advised by Reviewer 46id.
2. We provide more detailed clarifications and annotations regarding HR and TR in the paper. Advised by Reviewer 6jTA.
3. We add an explanation of “global structural consistency” in Section 1 (Introduction). Advised by Reviewer urNj.

> **Further Module Analysis**

1. We add an analysis of the PFSA computation paradigms (symmetric and asymmetric) in Appendix F. Advised by Reviewer 46id.
2. We add a description in Section 3.2 detailing the role of $(1-\gamma)\boldsymbol{z}$ in the high-dimensional manifold space. Advised by Reviewer 46id.
3. We include an ablation study on the annealing strategy of Attention Guidance in Appendix F. Advised by Reviewer 46id.
4. We provide a further analysis of the negligible encoding and decoding time cost of RepLDM in Appendix H. Advised by Reviewer 6jTA.
5. We provide a more detailed analysis of the role of PFSA in the latent space in Appendix A. Advised by Reviewer 6jTA and urNj.
6. We further discuss the performance of PFSA in both qualitative and quantitative experiments in Section 5.1. Advised by Reviewer 6jTA.
7. We analyze the computational complexity of Attention Guidance in Section 4.2. Advised by Reviewer ZoSx.
8. We add attention map visualizations of PFSA in Appendix A to provide a more intuitive explanation of its role. Advised by Reviewer urNj.

> **Further Comparison Experiments**

1. We add comparative experimental results and analyses using more appropriate metrics on the LAION dataset in Appendix J, and double the number of $4096\times 4096$ resolution samples to 2000. Advised by Reviewer 6jTA.
2. We repeat the experiments in the paper (including the newly added experiments on the LAION dataset) using a set of random seeds and compute the standard deviations to provide more robust evaluation results. Advised by Reviewer ZoSx.
3. We add experiments in Section 4.2 that evaluate model efficiency using NFE and FLOPs. Advised by Reviewer urNj.

---

### Decision · Program_Chairs · 2025-09-17

**Decision:**

Accept (spotlight)

**Comment:**

All four reviewers recommend acceptance. The paper introduces a novel and efficient framework for high-resolution image generation from pretrained latent diffusion models. The work's central innovation is a novel, parameter-free self-attention mechanism which is both elegant and able to enforce structural consistency without retraining. Its logical two-stage design first learns the image's overall structure before progressively adding detail in pixel space, managing to avoid common artifacts. The proposed method results in significant improvements in both image quality and inference speed, outperforming state-of-the-art techniques with almost 5x speedup. The paper's impactful solution, clever design, and broad applicability make it a strong contribution that's worth spotlighting.